# Structure–Activity Relationships in NHC–Silver Complexes as Antimicrobial Agents

**DOI:** 10.3390/molecules28114435

**Published:** 2023-05-30

**Authors:** Luisa Ronga, Mario Varcamonti, Diego Tesauro

**Affiliations:** 1Institut des Sciences Analytiques et de Physico-Chimie Pour l’Environnement et les Matériaux, Université de Pau et des Pays de l’Adour, E2S UPPA, CNRS, IPREM, 64053 Pau, France; luisa.ronga@univ-pau.fr; 2Department of Biology, University of Naples “Federico II”, Via Cynthia, 80143 Naples, Italy; mario.varcamonti@unina.it; 3Department of Pharmacy and Interuniversity Research Centre on Bioactive Peptides (CIRPeB), University of Naples “Federico II”, Via Montesano, 49, 80131 Naples, Italy

**Keywords:** NHC silver complexes, NHCs properties, structure activity relationships (SAR), anti-bacterial compounds

## Abstract

Silver has a long history of antimicrobial activity and received an increasing interest in last decades owing to the rise in antimicrobial resistance. The major drawback is the limited duration of its antimicrobial activity. The broad-spectrum silver containing antimicrobial agents are well represented by N-heterocyclic carbenes (NHCs) silver complexes. Due to their stability, this class of complexes can release the active Ag^+^ cations in prolonged time. Moreover, the properties of NHC can be tuned introducing alkyl moieties on N-heterocycle to provide a range of versatile structures with different stability and lipophilicity. This review presents designed Ag complexes and their biological activity against Gram-positive, Gram-negative bacteria and fungal strains. In particular, the structure–activity relationships underlining the major requirements to increase the capability to induce microorganism death are highlighted here. Moreover, some examples of encapsulation of silver–NHC complexes in polymer-based supramolecular aggregates are reported. The targeted delivery of silver complexes to the infected sites will be the most promising goal for the future.

## 1. Introduction

The fight against infections is one of the principal issues in medicinal care since the ancient eras. Although up to the XIX century microbes were not known the humankind used substances to prevent the effects of bacterial growth on food and wounds. In pharmacopeia, late-transition metal-based drugs have been played a crucial role. In old civilizations, the use of silver pots was largely diffused to avoid microbial contamination, to purify and store drinking water [1]. Metallic silver actually causes little damage to living organisms. Nevertheless, Ag^+^ is the biologically active species in applications of silver-containing compounds and formulations [2]. Von Nägeli reported that 10^−5^ to 10^−8^ molL^−1^ of Ag^+^ derived from metallic silver were effective to block the growth of *A. niger* spores [3].

Silver nitrate was used in the early 1800s for the treatment of ulcers, as antiseptic in wound care and to prevent eye infections in babies [4]. The irritation and the low stability are major problems of this salt. To overcome these drawbacks, colloidal silver solutions were introduced at begin of the 20th century able to release silver cations. After the Second World War, the discovery of penicillin with its strong antimicrobial properties led to decreased interest in silver compounds. A resumption of use did not occur until the 1960s when the discovery of silver sulfadiazine allowed treating with efficacy burn wounds combining the antibiotic properties of sulfonamide with silver [5]. Silver sulfadiazine is effective against a broad range of Gram-positive and Gram-negative bacteria, and its formulation is commercialized as topical antimicrobial cream the Silvadene^®^. The revival of silver compounds led to an increased attention of researchers to develop new compounds able to generate silver cations.

The activity of silver cations depends on their bioavailability and the delivery way [6]. It is increased by several parameters such as the presence of anions (i.e., sulfides, phosphates, chlorides) or cations (i.e., calcium and magnesium) high temperature and basic pH. For example, it was found that high concentrations of chloride anions increase the Ag^+^ cation concentration; indeed, in this condition, soluble AgCl_2_^−^ anionic specie is formed rather than solid AgCl [7]. Several other factors influence the antimicrobial and antifungal activity of silver salts [8]. The mechanism of action of Ag^+^ is not yet completely explicated. Silver cations target multiple sites on or within the bacterial cell. Four mechanisms have been identified as reviewed [9,10]. First, silver exhibits a strong tendency to be adsorbed to bacterial cell membranes interacting with proteins involved in cell wall synthesis as observed for the pathogen fungus *C. albicans* [11]. Inside the cell, silver cations can bind the DNA, they are able to interact with enzymes and membrane proteins and produce reactive oxygen species (ROS). Most of these mechanisms were identified detecting an increase in hydroxyl radicals generated by silver cations that cause the release of Fe^2+^ from FeS clusters [12]. Moreover, transmission electron microscopy (TEM) demonstrates morphological changes in the cell envelope. The interaction of silver ions with bacterial inner membrane is one of the most important mechanisms of Ag^+^ toxicity. Jung et al. [13] proved that the accumulation of Ag^+^ in the bacterial cell envelope is followed by the separation of the cytoplasmic membrane from the cell wall in both Gram-positive and Gram-negative bacteria.

Although the mechanism of damage of cell wall is still not fully understood, several hypotheses are reported in the literature. Silver ions may cause the formation of irregularly shaped pits in the outer membrane and change membrane permeability, which is caused by the progressive release of lipopolysaccharide molecules and membrane proteins. Another mechanism of bactericidal action may be based on the inhibition of cell wall synthesis interfering with the folding of involving enzymes.

Indeed, the latter mechanism can be induced by Ag^+^ binding thiol of the side chain of cysteine residues due to its soft acid properties. Therefore, many protein targets have been identified. Indeed, in 2019 Wang et al. recognized 34 proteins from *E. coli* that directly bind silver using liquid chromatography (LC) combined with gel electrophoresis (GE) and inductively coupled plasma mass spectrometry (ICP-MS) [14]. Among all the metabolic ways, the glycolysis was individuated as major pathway affected leading to depleted ATP. Moreover, Ag cations can affect cell respiration, inactivating NADH and succinate dehydrogenase, transport mechanism and metabolism increasing the ROS species [15]. The crystallographic structure of urease demonstrated that the His-Cys-Met sequence in the active site binds a bimetallic cluster of Ag cation affecting a protein loop near deactivating the enzymatic function [16]. These multifaceted modes of action of silver ions justify the scarceness of reports clearly explaining the silver resistance despite its diffused use. Another point of advantage of silver ions is their broad-spectrum antibiotic activity. Indeed silver, unlike conventional organic-based antibiotics, is active against a wide range of Gram-positive and Gram-negative bacteria.

However, the activity of silver salts has a limited time of action due to their low stability; therefore, the silver complexes can overcome this drawback releasing slowly the Ag^+^ cation. The choice of the ligands covalently bound to the metal center is pivotal to define the outcome of the Ag complex both in terms of its stability and in terms of its delivery on the site of action. Generally, silver(I)-S complexes show a narrower spectrum of antimicrobial activities than silver(I)-N and silver(I)-O complexes. Moreover, most of the investigated complexes with silver(I)-P bonds did not display antimicrobial activity. This evidence can be related to the high stability of their coordinative bonds and prompted researchers to explore the antibacterial properties of alternative silver ligands: i.e., N-heterocyclic carbenes (NHC). Since their discovery by Arduengo et al. [17] NHC have become universal ligands in organometallic and inorganic coordination to transition metals in catalytic applications [18] for his versatile properties. More recently, NHC ligands have been used as carrier molecules for metals in biological applications.

The goal of this review is to report about NHC–silver complexes tested in antimicrobial applications by highlighting their structure–activity relationships (SAR). For this purpose, the presentation of silver compounds is articulated here in four sections describing NHC-silver mononuclear, binuclear and loaded on supramolecular aggregates. In particular, the section NHC–silver mononuclear complexes presents different subparagraphs detailing the design of the structures of NHCs able to tune antimicrobial activity.

## 2. N-Heterocyclic Carbenes (NHC) Ligands

The properties of NHC ligands to coordinate metal centers were actively studied. They are strong nucleophiles and bind both main group and transition metals often with greater stability than phosphines [19]. Their structure is based on heterocycle containing at least one nitrogen atom contiguous carbene function. This class of ligands owns peculiar steric properties and strong σ donor of the carbene C2 due to the two adjacent nitrogen lone electron pairs to the free p-orbital of the carbon atom of NHCs (see Figure 1). This consequence reduces its π-backbonding capability, even stronger than alkyl phosphines [20].

Gusev quantified electron-donor and steric properties of a diverse group NHC with DFT calculations [21]. The σ-donor is strongly affected by substituents at C4 and C5 of the imidazole and imidazolidine rings, by the size of the N residues and by the NHC ring members (5 or 6 terms). The extended π-systems containing ligands are poor donors, whereas steric hindrance on sidechains of substituents on N do not affect bond energies. However, these substituents systematically can tune the lipophilicity.

In the literature, there are reported almost 20 types of ligands belonging to NHC class ranging from five- to seven-membered rings, but the most diffused types are: imidazolidin-2-ylidenes, imidazol-2-ylidenes and benzimidazol-2-ylidenes (see Figure 2).

Electronic and steric properties within these three types can be tuned inducing structural changes on the side chains of nitrogens. In medical applications these features are pivotal influencing the stability and the bioavailability. Among these ligands, the benzoimidazole is a vital pharmacophore of many biologically active heterocyclic compounds with a variety of pharmacological properties, i.e., anticancer, antiviral, antibacterial, antifungal, antihelminthic, anti-inflammatory and antihistaminic [22]. Indeed, benzimidazole derivatives are used in clinical applications such as anthelmintic (albendazole, mebendazole), fungicide (benomyl, carbendazim) and antacid (omeprazole, lansoprazole) this also motivate to investigate this type of molecular scaffold. The benzimidazole fragment and some of its derivatives suppress the bacterial growth which has been explained by their competition with purines resulting in inhibition of the synthesis of microbial nucleic acids and proteins [23].

These actions are related to the structure of benzimidazoles that allows them to form strong hydrogen bonds with enzymes and biological receptors, as well as participate in hydrophobic and π–π interactions, making them ligands for a variety of metal ions.

## 3. NHC–Silver Mononuclear Complexes

The binding interactions between the NHCs and the silver ion produce very strong bonds. Theoretical studies on silver chlorides demonstrated that these bonds are largely attributed to Coulombic attraction between the lone-pair electrons at the ligand donor atom and the positively charged metal atoms, but covalent interactions are not negligible. The covalent contribution growths from donation of the donor lone-pair electrons to the M-Cl σ* orbital. Moreover, the benzimidazolium core-based Ag(I)–NHC can be stabilized additionally by ring π–π interactions due to the extended π-cloud system. Ag(I)–NHC complexes have been observed to be very stable to air and moisture.

Ag(I)–NHC complexes are easily synthesized in situ by deprotonation of from azolium salts (e.g., imidazole, imidazoline, benzimidazole) with basic silver precursors such as Ag_2_O, Ag_2_CO_3_ or AgOAc. The Ag_2_O favors the formation of mono NHC or bis NHC by reacting with azolium whose positive charge is neutralized by halides or non-coordinate anions respectively.

The easy synthesis of Ag(I)–NHC has been widely exploited in the context of metal transferring hence relatively few publications on silver–carbene complexes are centered on their medicinal applications.

Youngs et al. reported the first studies in 2004 [24]. In the last two decades functionalized and nonfunctionalized NHC silver complexes have been extensively designed and synthesized for biological applications. Almost hundreds of complexes were reviewed for medical applications by Patil et al. and the Young [25,26]. In this review, NHC–silver mononuclear complexes are reported in subsections on the basis of the properties of the NHC ligands. The structures of the complexes described are reported in Figure 3, Figure 4, Figure 5, Figure 6, Figure 7, Figure 8, Figure 9, Figure 10 and Figure 11, whereas for each study the biological data of the most active complex are in the Table 1.

Roland et al. [27] carried out one of the first attempts to study antibacterial properties of a large panel of Ag(I)-NHC. They described 14 complexes designed for their catalytic properties but also with relevant antimicrobial properties. Significant changes in the activity were observed with slight differences in the NHC ligand structures. Ten of these complexes (**1**–**3** and **6**–**12**) (were found to display significant activity against *E. coli* with minimum inhibitory concentrations (MIC) ranging from 4 to 16 mgmL^−1^. Complexes **13** and **14** were the most active with MIC values of 1–4 mgmL^−1^, inhibiting both sensitive and resistant strains of *S. aureus* at clinically achievable concentrations. Moreover, Ag(I)–NHC s were able to reestablish the activity of Ciprofloxacin (CIP) by combining its synergistic effects.

**Figure 3 molecules-28-04435-f003:**
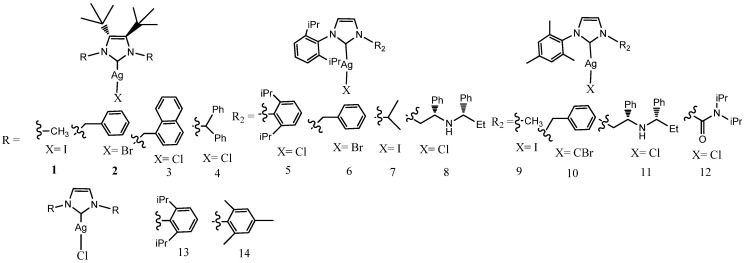
Structures of mononuclear NHC–silver complexes.

### 3.1. Benzyl-Substituted NHC–Silver Complexes

Because *p*-methoxybenzyl-substitution was successful applied in the case of the metallocene potential drugs, Tacke and coworkers designed six *p*-methoxybenzyl-substituted and benzyl-substituted Ag(I)–NHC acetate (Ac) derivatives (**15**–**20**) [28]. The structures of these complexes, like those of the other complexes described in this subsection, are shown in Figure 4.

To assess the antibacterial activity of compounds against both Gram-positive bacteria *Staphylococcus aureus* and Gram-negative bacteria *E. coli*, the Kirby–Bauer disk-diffusion method was applied. All imidazol-2-ylidene-Ag(I) complexes showed high antibacterial activity compared to the azolium salts and benzimidazol-2-ylidene-Ag(I) complexes that are poorly soluble. Applying the same methods in a later research study, six symmetrically substituted and non-symmetrically *p*-cyanobenzyl or benzyl substituted Ag(I)–NHC acetate derivatives (**21**–**26**) were tested [29].

Almost all complexes have displayed high antimicrobial activity with areas of clearance ranging from 4 to 12 mm, compared with the precursors and a marked improvement with respect to the compounds previously reported. Encouraging results pushed to study this class of compound designing other six non-symmetrically substituted and symmetrically substituted Ag(I)–NHC acetate derivatives (**27**–**32**) [30]. With respect to previously tested compounds, the concentration of stock solutions was reduced four-fold, and the Ag(I)–NHC complexes exhibited enhanced antibacterial activity. This achievement was due to the synergistic effects of the increased lipophilicity of the complexes. Chelation decreases the polarity of the metal ion, which further leads to the enhanced lipophilicity of the complex. The best antimicrobial activity was observed for the compound [1,3-dibenzyl-4,5-diphenylimidazol-2-ylidene]Ag(I) acetate (**32**). This complex was selected as the lead compound to carry out other assays on bacterial strains and to find the MIC [31]. The values of the MIC were observed to fall in the range 20 and 3.13 μgmL^−1^ after an incubation period of 20 h against the full bacterial panel. The activity was especially relevant against methicillin-resistant *S. aureus* (MRSA) with a MIC value of 12.5 μgmL^−1^. MRSA has acquired resistance to almost all β-lactam antibiotics including methicillin, amoxicillin and oxacillin among others. Some years later it was demonstrated that this complex at concentration of 25 μgmL^−1^ inhibited in vitro the growth of *S. aureus* by 71.2% and *C. albicans* by 86.2% [32]. Then its antibacterial property was evaluated in vivo in larvae of Galleria mellonella. Larvae inoculated with *S. aureus* or *C. albicans* exhibited increased survival after administration of **32** (20 μL at concentration 100μg.mL^−1^). Moreover, after the administration of **32**, the insect immune response was not observed. The lipid nature of the complex may not provoke an immune response as it may be better tolerated by the insect’s immune system. As consequence, it can be concluded that the increased survival of larvae that received **32** is due to the anti-microbial properties of the compound and not to a non-specific immune response induced by the introduction of the compound. This is the first demonstration of the in vivo activity against *S. aureus* and *C. albicans* without stimulation of non-specific immune response in larvae. Recently label-free quantitative proteomics was employed to analyze changes in protein abundance in the pathogenic yeast *C. parapsilosis* in response to treatment with **32** [33]. An increased abundance of proteins associated with detoxification and drug efflux were indicative of a cell stress response, whilst significant decreases in proteins required for protein and amino acid biosynthesis offer potential insight into the growth-inhibitory mechanisms of **32**. Achievements of proteomic findings, the prolific biofilm and adherence capabilities of *C. parapsilosis*, have demonstrated the potential of **32** to reduce epithelial cell adherence and biofilm formation and thereby decrease fungal virulence. They later investigated proteomic responses in the inhibition of growth to exposure of *S. aureus* and *P. aeruginosa* to **32** [34]. Both bacteria showed alterations in the abundance of proteins associated with the cell wall or envelope. However, in *P. aeruginosa*, a multitude of pathways was affected, including alginate biosynthesis, secretion systems, drug detoxification and anaerobic respiration. This contrasted with the response of *S. aureus*, where pathways such as protein synthesis, glucose metabolism and cell redox homeostasis were affected.

The relevant results pushed the same group to design and synthesize *p*-benzyl-substituted Ag(I)–NHC acetate compounds derived from 4,5-di-*p*-diisopropylphenyl or 4,5-di-*p*-chlorophenyl-1*H*-imidazole, (**33**–**42**) [35]. The complexes **34**–**37** were less active against both Gram-positive bacteria *S. aureus* and Gram-negative bacteria *E. coli* while **33**, **38**–**41** and **42** show moderate activity against *S. aureus*. However, their activity on *E. coli* remains considerably low. The introduction of isopropyl group and chlorine atom on benzoimidazole rings does not improve the antimicrobial properties observing weaker antimicrobial properties than the lead compound **32**.

Some years later Tacke et al. prepared four Ag(I)–NHC complexes substituting the acetate of the lead compound **32** with 4 substituted benzoates **43**–**46** [36]. These substitutions allowed the study of the effect that other ligands have on their antimicrobial activity. For instance, compounds **43**–**46** were screened for in vitro activity against two pathogenic bacterial strains, methicillin-resistant *S. aureus* and *E. coli*, and two fungal strains, *C. albicans* and *C. parapsilosis*. The results displayed a clear effect on the modification, with fluorine atom bearing on benzoate compound **46** showing the best inhibition against *C. parapsilosis*.

In 2017, Bhagat et al. reported in two different manuscript the synthesis of seven neutral bromide Ag(I)–NHC complexes and seven cationic NHC–Ag(I)–NHC complexes [36,37] Neutral and cationic species formation is influenced by modulating the polarity of the solvent from dichloromethane to methanol. All complexes were evaluated against one Gram-negative (*S. enterica*) and one Gram-positive (*S. aureus*). In the first paper the most potent compounds **47** and **51** inhibited the microbial growth with MIC value of 6.25 μM against *S. aureus* and **47** displayed MIC of 25 μM against *S. enterica* showing an efficient growth inhibition even up to 21 d. The treatment of bacteria with **47** and **51**, and examining cell wall appearance under SEM infers the cell wall disrupting effects of silver complexes. The factors such as hydrophobic substitutions and steric bulk affect lability of Ag-Carbene bond and thereby regulating the release of Ag^+^ in an aqueous environment. The positive charge seem not to influence the silver activity in this case. In the second paper [38], the best inhibition among the six complexes, was detected with the nitrobenzyl and biphenyl carbonitrile based neutral silver **57** complex displaying MIC of 25 μM in *S. enterica* as well as in *S. aureus*. The silver complexes had higher MIC in *S. enterica* because of the resistance for silver ions. The other factors such as hydrophobic substitution and steric bulk are involved in the release of Ag^+^ cations.

At the end of the last decade several groups studied the influence of substituents on benzyl moiety on imidazole ring. For instance, Haque et al. studied non-symmetrical six NHC–Ag(I)–NHC cationic complexes, **62**–**67** to evaluate the effect of electron-withdrawal or electron-donors on benzyl ring [39]. They introduced nitrile groups on different positions of an *N*-benzyl ring to compare the antibacterial effects with methyl groups positioned in same positions. The substitution does not induce a relevant difference in antimicrobial properties. All six complexes displayed a modest to good inhibition zone against both *E. coli* and *S. aureus*, around 9 mm with concentration at 6 μL. Very recently Sahin et al. reported the synthesis of five neutral Ag(I)–NHC complexes bearing on imidazole benzyl ring para substituted with electron withdrawal or donor groups (**68**–**72**) [40]. All compounds were also analyzed by molecular docking methods with the potential target molecules such as N-Acyl Homoserine Lactone (AHL) Lactonase. All compounds have a high antimicrobial activity against Gram-positive and Gram-negative microorganisms but displayed the greatest effect on *C. albicans* fungi. MICs of all complexes were in the range 18–36 μM. The best inhibitor of *E. coli* biofilms was found the compound bearing the benzyl (**68**), while the bromide complex (**72**) inhibited *C. albicans* biofilm at the highest rate. In general, all reported studies indicate that the presence in different position of substituents on benzyl groups does not affect significantly the antimicrobial efficacy.

**Figure 4 molecules-28-04435-f004:**
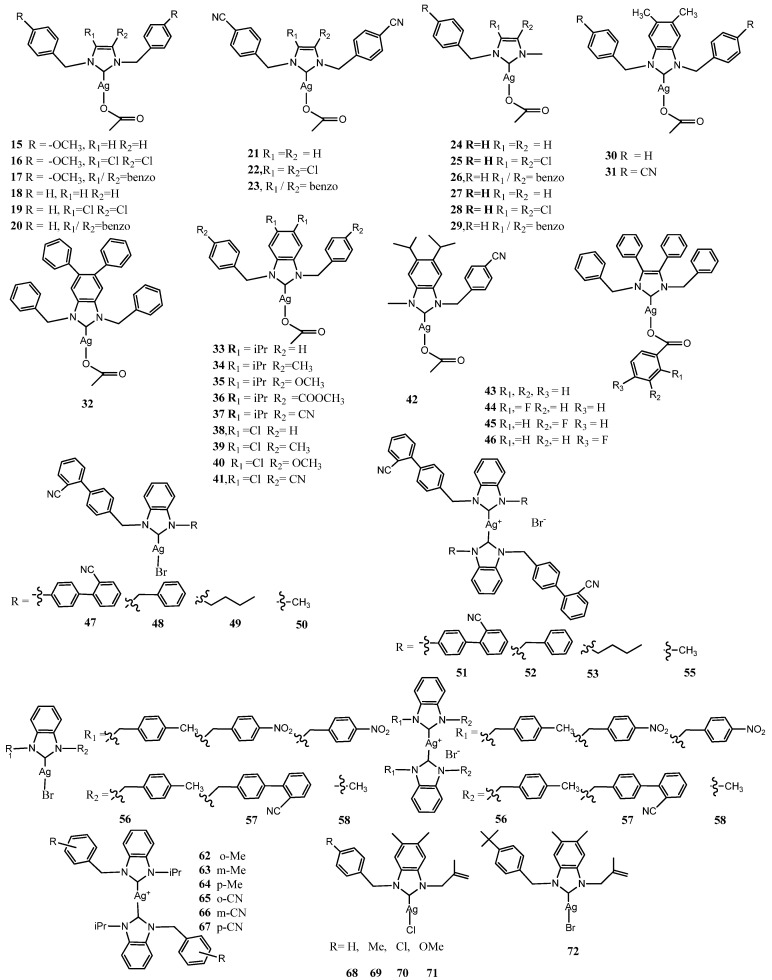
Structures of benzyl-substituted NHC–silver complexes.

### 3.2. NHC–Silver Complexes Designed with Steric Hindrance

Many attempts have been devoted to study the impact of bulky substituents on imidazole group on antimicrobial activity; the structures are reported in Figure 5. Özdemir and coworkers synthesized 10 bulky 3,5-di-*tert*-butyl substituted NHC–Ag-Br complexes (**73**–**82**) [41]. The evaluation was carried out against two Gram-negative bacteria (*E. coli* and *P. aeruginosa*), two Gram-positive bacteria (*S. aureus* and *E. Faecalis*) and two fungal strains (*C. albicans* and *C. tropicalis*). Reduced hindrance containing complexes (such as **81** and **82**) have less antimicrobial activity than the bulky complexes. Indeed, the bulkiest complex **77** was the most effective against all the tested microbial strains, with a MIC of 6.25 μgmL^−1^, whereas smaller substituents bearing complex **81** exhibited an MIC from 25 to 100 μgmL^−1^.

The influence of steric hindrance was previously studied introducing polymetyl-aryl group in seven 4-vinylbenzyl-NHC silver complexes (**83**–**88**) [42]. The phenyl and polymethyl aryl substituted complexes **83**, **86** and **87** showed good antibacterial activity, especially against *C tropicalis* and *C albicans* as the fungal strains. The same compounds acted in similar way against Gram-negative (*E. coli*, *P. aeruginosa*) and Gram-positive (*E. faecalis*, *S. aureus*) bacterial strains. The monosubstituted metylaryl Ag(I)-NHC complexes **84**, **85** were less active. The same group later expanded the series of the 1-(4-vinylbenzyl)benzimidazole NHC–Ag(I)-Cl complexes synthesizing alkyl functionalized ligands as well as aryl ligands (**89**–**95**) [43]. The antimicrobial activity was evaluated against three Gram-positive bacterial strains (*B. subtilis*, *L.monocytogenes*, *S. aureus*), seven Gram-negative bacterial strains (*E.coli*, *K. pneumoniae*, *P. mirabilis*, *P. aeruginosa*, *S. typhimurium*, *Y. enterocolitica*) and one yeast (*C. albicans*) by agar diffusion assay. In this case alkyl and aryl bearing substituents on Ag(I)–NHC complexes (**89**–**95**) exhibited similar inhibition zone (ZoI) between 13–19 mm without detecting very significant differences between the substituents.

**Figure 5 molecules-28-04435-f005:**
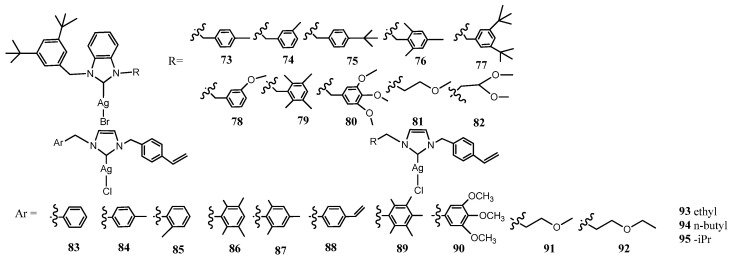
Structures of NHC–silver complexes designed with Steric Hindrance.

### 3.3. NHC–Silver Complexes Bearing p–Nitrobenzyl Group

Patil et al. studied series of NHC functionalized with *p*–nitrobenzyl (Figure 6). This moiety was selected due the relevance of the nitrobenzene derivatives, which are being used in the synthesis of many drugs acting as analgesics, antipyretics and antipsychotic. The authors reported the synthesis of eight cationic NHC–Ag–NHC complexes (**96**–**103**), from non-symmetrically *p*-nitrobenzyl- and *p*-cyanobenzyl-substituted salts [44]. The isopropyl (**96**) and *p*-cyanobenzyl-substituted (**100**–**103**) showed in the evaluation antimicrobial activity the best activity against Gram-negative (*E. coli*), whereas *i*butyl (**96**) and *p*-cyanobenzyl-substituted (**100**–**103**) were the most effective against Gram-positive (*S. aureus*).

Later they studied the effect of substituents on position 4 and 5 on imidazole rings synthetizing four non-symmetrically *p*-nitrobenzyl-substituted Ag(I)–NHC acetate (**104**–**107**) and four respective cationic NHC-Ag(I)–NHC complexes (**108**–**111**) [45]. In general, all of the complexes displayed MIC ranging between 8.0 and 128 μgmL^−1^ in vitro antibacterial activity against two Gram-positive bacteria (*S. aureus* and *B. subtitis*) and four Gram-negative bacteria (*E. coli*, *P. aeruginosa*, *S. sonnei* and *S. typhi*). The more active among the Ag(I)–NHC acetate was **107** complexes with highest bioactivity against *S. aureus* and *E. coli* (MIC 8 μgmL^−1^) whereas the more active among the NHC–Ag(I)–NHC complex was 4,5 diphenyl imidazole- Ag(I)–NHC (**110**) with the highest bioactivity against *E. coli* (MIC 8 μgmL^−1^).

The same group followed their studies on *p*-nitrobenzyl-substituted NHC bearing alkyl chain with different length on imizole ring. They prepared five Ag– NHC acetate complexes (**112**–**116**) and five NHC–Ag(I)–NHC complexes (**117**–**121**) [46]. All these complexes were screened against the same bacterial strains. The antibacterial activity of Ag(I)-NHC acetateAc and NHC–Ag(I)–NHC complexes was comparable with the MIC ranging between 16.0 and 128 μgmL^−1^, observing that the MIC decreases with an increase in the alkyl chain length. The increase in antibacterial activity was already detected by Asekunowo et al. in previous paper [47]. They compared three types of nonsymmetrically alkyl substituted *N*-benzyl benzimidazole NHC–Ag(I)–NHC complexes (**122**–**124**). The effect of the N-alkyl substitution on antibacterial activities was evaluated against two bacteria strains, the Gram-positive *S. aureus* and the Gram-negative *E. coli* with a MIC value ranging between 12.5–50 μgmL^−1^. The complexes bearing on imidazole longer alkyl chain length displayed major activity against both bacteria strains. This suggests that in this case ligands may simply facilitate the transport of Ag(I) ions to their biological targets. Moreover, all complexes were efficient in promoting cleavage or degradation of DNA.

More recently Granillo et al. synthesized [Ag(1–methyl–3–(4–nitrobenzyl)–1H–imidazoleCl]_2_ **125** [48]. The dimeric molecular structure of the complex was determined by single crystal X–ray diffraction. The antibacterial assays in vitro were performed against two Gram-positive (*S. aureus* and *B. subtilis*) and two Gram-negative species (*P. aeruginosa* and *E. coli*) by Kirby–Bauer’s disk diffusion method. Compound exhibited moderate activity against *S. aureus* and *P. aeruginosa* when 4.31 μg of Ag were used, similar behavior was found against *B. subtilis* and *E. coli* but only when 5.74 μg of Ag were employed. The ZoI tested against *S. aureus* is comparable to other products previously reported with *p*-nitrobenzyl-substituted benzoimidazole Ag(I)-NHC **112**.

**Figure 6 molecules-28-04435-f006:**
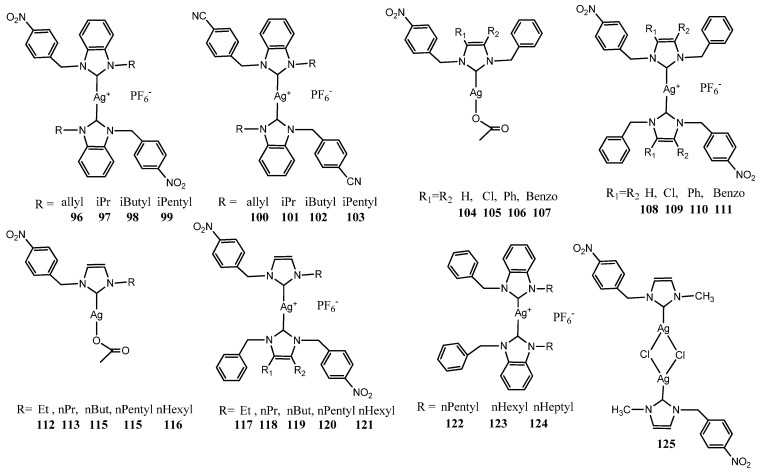
Structures of NHC–silver complexes bearing p–nitrobenzyl group and structures of compared complexes to them.

### 3.4. NHC–Silver Complexes Bearing Naphthalil Group

Ott et al. synthesized NHC–Ag acetate complexes bearing bis-1,8-naphthalimide ligands (**126**) (Figure 7). This complex combine the DNA interacting naphthalimide with a metal based mechanism of drug action [49]. The unsubstituted bis-naphthalimide Ag(I)–NHC complex **126** was evaluated against three Gram-negative bacteria (*E. coli*, *A. baumannii* and *P. aeruginosa*) and three Gram-positive bacteria (*B. subtitis* and two *S. aureus* strains, DSM 20231 and ATCC 43300). The growth of Gram-negative bacteria was not significantly affected while Gram-positive bacteria were more sensitive with the most relevant value of MIC (16 μgmL^−1^) for *B. subtitis*. This might be due to differences in the cell envelops of Gram-positive and Gram-negative bacteria, efflux systems, and/or to metal-based inhibition of bacterial thioredoxin reductase (TrxR), which is more harmful for Gram-positive bacteria [50]. Later Gök et al. reported the synthesis of seven naphthalen-1-ylmethyl substituted Ag(I)–NHC complexes (**127**–**133**) [51]. These NHC–Ag-Cl complexes were screened in vitro toward four bacteria (*S. aureus*, *E. coli*, *E. faecalis* and *P. aeruginosa*) and two fungi (*C albicans* and *C tropicalis*). This research confirmed the results obtained by Ott et al. [49] finding to be effective only against Gram-positive bacteria. Indeed complex **127** is the most effective compound against *S. aureus* with a MIC of 6.25 μgmL^−1^, and complex **128** displayed the highest activity against the fungal strains with a MIC of 6.25 μgmL^−1^.

**Figure 7 molecules-28-04435-f007:**
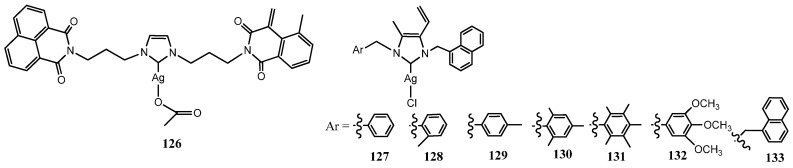
Structures of NHC–silver complexes bearing naphthalil group.

### 3.5. NHC–Silver Complexes Bearing Electron Withdrawing Substituents

Stability of Ag(I)–NHC complexes can be improved by the use of electron withdrawing groups at the 4 and 5 positions of the imidazolin-2-ylidene ligand (Figure 8). Youngs and co-workers designed three Ag(I)-NHC acetate complexes based on 4,5,6,7-tetrachlorobenzimidazole (**134**–**136**) [52]. The benzimidazole allows to add two more of the sigma-withdrawing π-donating chlorine atoms than imidazole increasing the stability of the NHCs. These complexes proved highly efficacious with MICs ranging from 0.25 to 6 μg mL^−1^. The addition of a hydroxyethyl substituent in **135** increase the relative water solubility. It was the best inhibitor on growth of the silver-resistant J53 + pMG101 *E. coli* strain (MIC 8 μL) while the methylnaphthyl bearing complex **136** exhibited bactericidal activity against silver-resistant bacteria. Asekunomo et al. synthesized three NHC–Ag–NHC complexes containing either an alkyl nitrile or an aryl nitrile. (**137**–**139**) [53]. They detected that all complexes could effectively bind DNA. Furthermore, the complexes **137**–**139** in vitro were evaluated against both *E. coli* and *S. aureus* showing moderate antibacterial activity without significantly difference. Very recently, three mono butyronitrile funtionalized benzimidazole Ag(I)–NHC complexes were synthesized by Turker et al. (**140**–**142**) [54]. Although their antibacterial activities are moderate, they are more active higher against Gram-positive (*S. aureus*, *methicillin-resistant S aureus* and *E faecalis*) strains with MIC in ranging 12.5–100 μgmL^−1^ for **142** than against Gram-negative (*E. coli*, *K. pneumoniae*, *A. baumannii* and *P. aeruginosa*) strains. Moreover, promising results were achieved against the standard fungal *C albicans* and *C. glabrata*.

Carbonyl groups were introduced by Haque et al. that designed two Ag(I)–NHC acetate and NHC-Ag(I)–NHC complexes **143**–**144** bearing a keto alkyl chain to study the effect of carbonyl group against the proliferation of bacteria [55]. Biological assays results indicate that both compounds possess a good antimicrobial activity against both *E. coli* and *S. aureus*, with an MIC of 31.25 μgmL^−1^. Furthermore, all tested compounds cleaved DNA.

Kaloğlu et al. reported the synthesis of eight substituted *N*-(2-(2-ethoxy)phenoxyethyl)benzimidazole Ag(I)-NHC complexes (**145**–**152**) [56]. These complexes were studied at different temperature by NMR. These investigations have furnished evidence for ligand-exchange equilibria between neutral monocarbene complexes [AgX(NHC)] and ion pairs [Ag(NHC)_2_][AgX_2_] (X = halogen) through an associative mechanism. All complexes exhibited good activities against standard bacterial strains (*E. faecalis*, *S. aureus*, *E. coli* and *P. aeruginosa*) and the fungal strains (*C. albicans* and *C. tropicalis*). However, the most lipophilic complex (**151**) was found out as the most active especially against Gram-positive and fungal strains with a MIC of 6.25 μgmL^−1^. Other most lipophilic complexes **145 148** were active with an effective concentration range of 6.25–25 μgmL^−1^.

**Figure 8 molecules-28-04435-f008:**
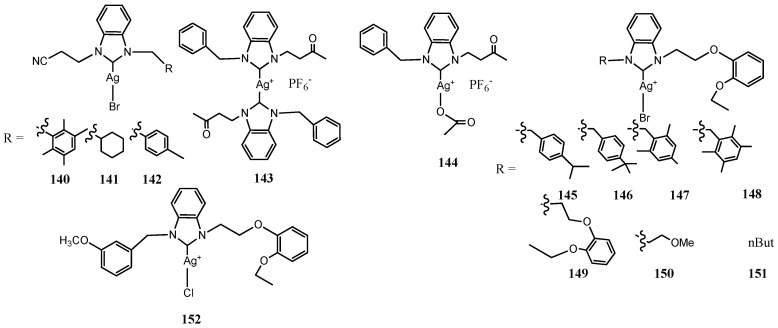
Structures of NHC–silver complexes bearing electron withdrawing substituents.

### 3.6. NHC–Silver Complexes Bearing Aliphatic Substituents

The relevance of lipophilic groups to favor antimicrobial activity pushed to synthesize a series of highly lipophilic 1-(2-Cycloheptylethyl substituted Ag(I)–NHC complexes bearing aryl and anthracene moieties (**153**–**156**) (Figure 9) [57]. All compounds performed similar antibacterial and antifungal properties compared to those found in the previous study (MIC: 6.25 μgmL^−1^ for Gram-positive bacteria and fungi strains).

Four allyl substituted Ag(I)–NHC complexes bearing polimethyl-aryl moieties were synthesized by Üstün et al. (**157**–**160**) [58]. Silver complexes showed higher antimicrobial activity against all microorganisms tested and better antifilm properties than the corresponding salts. Specifically, the **160** complex exhibited anticandidal activity similar to that of Flucanozol and reduced significantly *E. coli* and *C. albicans* biofilm zone. The best behavior of **160** can be attributed to the increase in the number of alkyl groups on benzyl moiety that increases in lipophilicity. Later Tutar et al. expanded the investigation of this class of compounds synthesizing other six allyl-substituted benzimidazole-based Ag(I)–NHC complexes (**161**–**166**) [59]. All tested compounds showed strong and similar activity against *E. coli* ATCC 25922 at very low concentration (≤3.9 μgmL^−1^) as compared to ampicillin as well as they displayed the same or stronger activity against the *A. baumannii* isolate with MIC values of 15.6–31.25 μg mL^−1^. The authors claim more efficacies of these compounds versus same strains used by Achar et al. [60]. All compounds inhibited the formation of *E. coli*, *K. pneumoniae*, *E. faecalis* and *A. baumannii* biofilms at sub-MIC concentrations in the range 30–90% and *C. albicans* biofilm formation by 38–53%. In the same year, a parallel paper reported other four 1-allyl 5,6-dimethylbenzimidazole Ag(I)–NHC complexes with methyl substituted aryls in different position compared to *t*but substituted (**167**–**170**) [61]. The best performance against all microorganisms was detected for **170** with MIC < 1.9 μgmL^−1^. Moreover, in this case NHC salts showed moderate antifungal and antibiofilm activity while the others silver complexes had anyway strong antimicrobial activity with MIC values <3.9–15.6 μgmL^−1^ without relevant difference changing the methyl position. In particular, all methyl functionalized benzoimidazole silver complex increases the activity against *C. albicans* (<1.9 μgmL^−1^) compared with a similar compound previously reported unfunctionalized benzoimidazole silver complex (7.8 μgmL^−1^).

**Figure 9 molecules-28-04435-f009:**
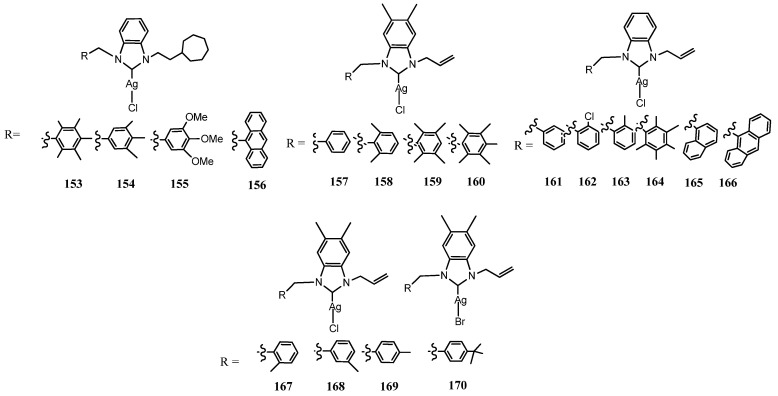
Structures of NHC–silver complexes bearing aliphatic substituents.

### 3.7. NHC–Silver Complexes with Nitrogenous Substituents

Mnasri et al. prepared silver complex functionalizing non-symmetrical benzoimidazole ring with diisopropylamino)ethyl or pyrrolidin-1-yl groups and the other nitrogen with a polymethyl substituted benzyl (**171**–**176**) (Figure 10) [62]. The authors designed these ligands to evaluate the effect of steric hindrance on Ag(I)-NHC complexes. It has been found that the complexes were antimicrobially active with MIC values between 0.24 and 62.5 μgmL^−1^ and showed higher activity than the free ligand. Results indicated that the presence of sterically bulky groups directly grafted on the nitrogen atom of the benzimidazol-2-ylidene ligand has a positive effect on the antimicrobial activity. In particular the silver complexes **173** and **171**, displayed inhibition of bacterial growth with MICs of 0.24 and 1.95 mgmL^−1^, respectively, against *L. monocytogenes* ATCC 19117 and *S. typhimurium* ATCC 14028.

Recently, Muniyappan et al. synthesized silver complexes bearing picolyl and benzyl linked biphenyl NHC ligands **(177**–**179**) [63]. These biphenyl NHC ligands showed the inherent sigma donating properties through proton-coupled carbon NMR spectroscopy. All silver complexes showed comparable antibiotic activity against *P. aeruginosa* as that of reported antibiotic lead compound **32** and enhanced compared to previous reported *i*butyl substituted benzimidazole Ag(I)–NHC complexes and to the reported phenyl-substituted NHC precursors. All complexes showed promising antimicrobial activity against Gram-positive (*S. aureus* IE903) and Gram-negative (*P. aeruginosa* E322) bacterial strains cultured from the human clinical isolates.

**Figure 10 molecules-28-04435-f010:**
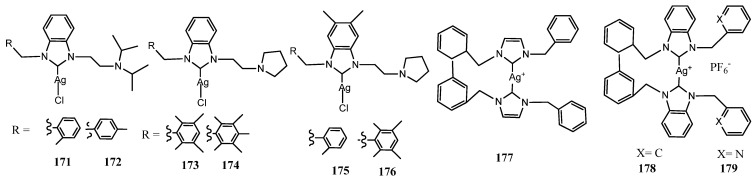
Structures of NHC–silver complexes with nitrogenous substituents.

### 3.8. Unusual NHC–Silver Complexes

Knowledges have been increased investigating non-classical NHC silver compounds in the last years. Very recently Sanchez et al. prepared {Ag[NHCMes,R]}_n_ polymeric complexes (**180**–**184**) (Figure 11) [64]. In the solid state, **180** is a one-dimensional coordination polymer, in which the Ag^+^ cation is bonded to the carbene ligand and to the carboxylate group of a symmetry-related Ag[NHCMes,H] moiety. Antimicrobial properties of these complexes were evaluated versus Gram-negative bacteria *E. coli* and *P. aeruginosa*. From observed MIC and Minimum Bactericidal Concentration (MBC) values, complex **182** showed the best antimicrobial properties (eutomer), which were significantly better than those of its enantiomeric derivative **182** (distomer). Additionally, analysis of MIC and MBC values of **182**–**184** reveals a clear structure–antimicrobial effect relationship. Antimicrobial activity decreases when the steric properties of the R alkyl group in {Ag[NHCMes,R]}_n_ increase.

Di Napoli et al. synthesized five silver complexes having bidentate NHC ligands **185**–**189** [65]. Four of these ligands were neutral, having an alcohol group on alkyl substituent of one of the two nitrogen atoms of the heterocycle [NHC–OH], the fifth, having a ligand alkoxide, is mono-anionic [NHC–O^−^]. All the synthesized complexes have a significant antibacterial activity against *E. coli* and *B. subtilis* (MIC 5–50 μgmL^−1^). Probably, the pincer effect of both [NHC–OH] and [NHC–O^−^] ligands stabilizes these compounds slowing the cation release.

Prencipe et al. tested acridine scaffold and detailed nonclassical pyrazole derived Ag(I)–NHC neutral (**190**, **192**–**193**) and NHC-Ag(I)–NHC cationic complexes (**191**, **194**) [66]. Their inhibitor effect on two Gram-negative bacteria: *E. coli DH5α* and *P. aeruginosa* PAOI and two Gram-positive bacteria: *S*. *aureus* 6538P and *B. subtilis* PY79 was evaluated. The acridine attached to imidazole core in neutral and cationic complexes has shown effectiveness at extremely low MIC values (less than ≤1 μM). Although pyrazole NHC silver complexes were less active than the acridine Ag(I)–NHC (≤50 μM), they represent the first example of this class of compounds with antimicrobial properties. Another unusual NHC ligands is derived from 1,2,4-triazoles. The presence of a third peripheral nitrogen of the heterocyclic ring induces significant differences compared to imidazole-based ligands. This modification causes dissymmetry, and reduces the strength of sigma donation whilst increasing the π-accepting properties of these ligands and increase the acidity of the pre-carbenic C5 proton.

Mather et al. synthesized a series of Ag(I) (**195**–**202**) complexes of 1,2,4-triazolylidene and imidazolylidene based NHC ligands [67].

The antibacterial activity has been evaluated providing a direct comparison between the influence of 1,2,4-triazolylidene versus imidazolylidene ligands on these properties. All complexes were not active against Gram-positive bacteria strains while they show antibacterial properties against Gram-negative. The lowest MIC values (2–4 μg mL^−1^ and 2 μg mL^−1^) were found for **200** and **201** against *A. baumannii* and the magnitude of these values were similar to the antibiotic colistin (≤2 μg mL^−1^).

The complexes bearing dimethyl and diethyl substituents **199**, **200** and **201** were found to have greater levels of antibacterial activity in comparison to the phenyl substituted complexes **196** and **197** while the 1,2,4-triazolyle and imidazole rings do induce clear and relevant difference. Complexes **200** was investigated to determine the propensity to develop bacterial resistance, and no resistance was observed for *A. baumannii*.

**Table 1 molecules-28-04435-t001:** Best NHC–silver inhibitors of each reference.

Entry	N° of Tested Strains	Highest Activity Against	Concentration	ZoI or MIC or CA *	Ref.
**1, 7**	5	*E. coli*	4 mg mL^−1^	MIC	[27]
**14**	5	*S. aureus*	1 mg mL^−1^	MIC	[27]
**32**	7	*M. smegmatis*	5 mg mL^−1^	MIC	[31]
**42**	2	*S. aureus*	7 mm	CA	[35]
**47, 51**	2	*S. aureus*	6.25 μM	MIC	[37]
**57**	2	*S. aureus*	25 μM	MIC	[38]
**62**	2	*S. aureus*	14 mm at 12 μL	ZoI	[39]
**72**	5	*E. faecalis*–*C. albicans*	14 μM	MIC	[40]
**77**	4	*All strains*	6.25 μg mL^−1^	MIC	[41]
**83, 86, 87**	6	*C. tropicali*–*C. albicans*	25 μg mL^−1^	MIC	[42]
**89**	11	*S. aureus*	19 mm at 50 μL	ZoI	[43]
**96**	2	*S. aureus*	17 mm at 2.85 nM	ZoI	[44]
**102**	2	*S. aureus*	17 mm at 2.85 nM	ZoI	[44]
**107**	6	*S. aureus*–*E. coli*	8 μgmL^−1^	MIC	[45]
**110**	6	*E. coli*	8 μgmL^−1^	MIC	[45]
**115**–**116**	6	*S. aureus*–*E. coli*	16.0 μgmL^−1^	MIC	[46]
**120**–**121**	6	*S. aureus*–*E. coli*	16.0 μgmL^−1^	MIC	[46]
**123**–**124**	2	*S. aureus*	12.5 μgmL^−1^	MIC	[47]
**125**	4	*S. aureus*–*P. aeruginosa*	12 mm at 9 μL	ZoI	[48]
**126**	6	*B. subtitis*	16.0 μgmL^−1^	MIC	[49]
**127**	6	*S. aureus*	6.25 μgmL^−1^	MIC	[51]
**128**	6	*C. albicans*–*C tropicalis*	6.25 μgmL^−1^	MIC	[51]
**135**	8	*E. coli J53* + *pMG101*	8 μgmL^−1^	MIC	[52]
**137**	2	*S. aureus*	13.5 mm at 25 μgmL^−1^	ZoI	[53]
**142**	9	*S. aureus*	12.5 μgmL^−1^	MIC	[54]
**143**–**144**	2	*S. aureus*–*E. coli*	31.25 μgmL^−1^	MIC	[55]
**151**	6	*E. faecalis*–*S. aureus*	6.25μgmL^−1^	MIC	[56]
**153**–**156**	6	*C. albicans*–*C tropicalis*	6.25 μgmL^−1^	MIC	[57]
**153**–**156**	6	*E. faecalis*–*S. aureus*	6.25 μgmL^−1^	MIC	[57]
**158, 160**	4	*E. faecalis*–*S. aureus*–*E. coli*	7.8 μgmL^−1^	MIC	[58]
**161**–**166**	6	*E. coli*	≤3.9 μgmL^−1^	MIC	[59]
**170**	5	*S. aureus*	<1.9 μgmL^−1^	MIC	[61]
**173**	6	*L. monocytogenes*	0.24 mgmL^−1^	MIC	[62]
**177**–**179**	3	*P. aeruginosa E322*	10 ppm	MIC	[63]
**182**	2	*P. aeruginosa*	39 μM	MIC	[64]
**188**–**189**	2	*E. coli*–*E.**subtitis*	5 μgmL^−1^	MIC	[65]
**192**–**194**	4	*All Strains*	<1μM	MIC	[66]
**200**–**201**	4	*A. baumannii*	2 μg mL^−1^	MIC	[67]

* ZoI (inhibition zone), CA (Clearance Area), MIC (minimum inhibitory concentrations).

**Figure 11 molecules-28-04435-f011:**
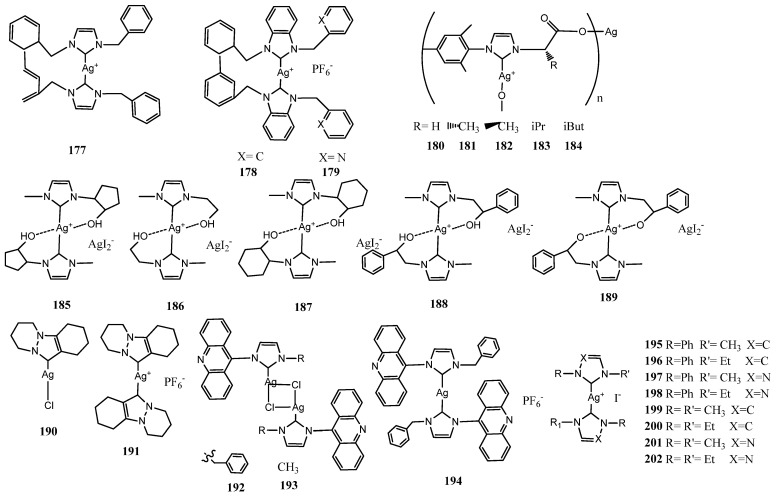
Structures of unusual NHC–silver complexes.

## 4. NHC–Silver Complex with Ligand in Dual Activity

A class of complexes was designed to exploit the synergic activity of silver cations and the ligand with antibiotic properties moiety. All structures of reviewed complexes in this section are in Figure 12 and the antimicrobial data of the most active inhibitor are in Table 2.

Caffeine is a xanthine derivative; it is a good candidate as carrier molecule for the delivery of silver cations to the lungs. Young and co-workers introduced a methyl group in the structure of imidazole moiety of caffeine to synthesize NHC [68]. This complex **203** exhibited high in vitro antimicrobial efficacy against a broad spectrum of highly resistant respiratory pathogens including members of the *Burkholderia cepacia* complex and against *E. coli* with MIC values as low as 1 μgmL^−1^. The addition of hydroxyl such as a hydroxyethyl group on methylcaffeine appears to greatly enhance the water solubility of the resulting silver carbene complex favoring the use in spray for future treatment of virulent and multidrug-resistant (MDR) pulmonary infection [87]. The coumarin derivative class is possible to evaluate for their antibiotic properties. Antimicrobial coumarin derivatives, such as novobiocin, clorobiocin and coumermycin A_1_, exert their antibacterial activity by inhibition of the type II DNA topoisomerase DNA gyrase [88]. Karatas et al. synthesized eight Ag(I) complexes coordinate by NHC ligands which tethered coumarin group (**204**–**211**) [69]. The cationic NHC–Ag–NHC complexes have a structure such as [AgL_2_]^+^[AgCl_2_]^−^. The salts and the silver complexes were evaluated against two Gram-negative bacteria (*E. coli* and *P. aeruginosa*), two Gram-positive bacteria (*S. aureus* and *E. faecalis*) and two fungal stains (*C. albicans* and *C. tropicalis*). The results of this study displayed that addition of coumarin to the NHC scaffold did not increase antibacterial and antifungal activities significantly. More relevant is the presence of naphthyl group (**210**) that induces the highest activity with an MIC of 25 μgmL^−1^ for Gram-positive and fungal strains. After this disheartening result Achar et al. studied the behavior of five 6-chlorocoumarin NHC–Ag–NHC complexes, with different *N*-alkyl groups (**212**–**216**) [70]. The complexes were evaluated against two Gram-positive bacteria (*S. aureus* and *B.subtitis*) and four Gram-negative bacteria (*E. coli*, *P. aeruginosa*, *S. typhi* and *S. sonnei*). Against these bacterial strains, salts did not show any activity with a MIC above 128μgmL^−1^. However, the corresponding silver complexes demonstrated moderate to high antibacterial activity in the range 7 ± 1–12 ± 1 and 7 ± 1–25 ± 2 mm diameter of zone of bacterial growth inhibition against *S. aureus* and *E. coli*, respectively. This activity of complexes in the case of *E. coli* is almost two-fold higher than the antibacterial potentials of benzimidazolium salts. The complex with an isopropyl moiety attached to imidazole **212** core exhibited the highest bioactivity against *P. aeruginosa* (MIC = 8 μgmL^−1^ and ZoI= 24 ± 1 mm). In a following study fourteen 6-methylcoumarin-substituted NHC–Ag–NHC (**217**–**223**) and 6-methylcoumarin-substituted Ag(I)-NHC acetate (**224**–**230**) complexes, with different *N*-alkyl groups attached to their imidazole core, were synthesized [71]. Tested against the same strains, all the complexes were found inactive in the case of *B. subtilis*, *S. typhi* and *S. sonnei*. However, it was found that methyl and butyl bearing imidazole NHC–Ag–NHC complexes (**217**, **220**) showed exceptional activity against *P. aeruginosa* with a MIC of 8 μgmL^−1^. This activity is comparable to that displayed by the 6-chlorocoumarin. Furthermore, no correlation could be drawn with increasing steric bulk on the NHC donors and the observed activity. However, NHC–Ag–NHC complexes displayed better results than their acetate counterparts. To explore the impact of different coumarin substituted NHC ligand backbones, the same authors synthesized silver complexes with a series of structurally related ether–functionalized imidazolium and benzimidazolium hexafluorophosphate salts bearing 6–methylcoumarin, 6–chlorocoumarin and 5,6–benzannulated coumarin substituents (**231**–**242**) [72]. In antibacterial evaluations, silver complexes displayed promising activity with the MIC values in the range 8–64 μgmL^−1^ against *S. aureus*, *B. subtilis*, *E. coli* and *S. typhi*, while the corresponding salts were almost no activity. Note that no appreciable differences were detected tuning the substituents on coumarin whereas generally the cationic complexes (**231**–**236**) were more active than Ag(I)–NHC complexes (**237**–**242**). The same group, following their studies on coumarin derivatives, prepared two different series of pro-ligands using 2–bromomethylbenzonitrile and substituted 4–bromomethylcoumarin derivatives to functionalize the imidazole and benzoimidazole ring. (**243**–**250**) [60]. The pro-ligands and the corresponding silver complexes were evaluated for antibacterial activity against Gram-positive (*S. aureus*) and Gram-negative bacteria (*E. coli*). In these series the salts were totally inactive against both the bacterial strains tested with the MIC being more than 128 μgmL^−1^, whereas corresponding Ag(I)–NHC complexes exhibited against *E. coli* very significantly value of MIC of 8 μgmL^−1^ except for 7,8 benzosubstituted. Conversely, benzimidazole derived silver complexes having methyl– and chloro–coumarin substituents displayed MIC of 32 μgmL^−1^ against *S. aureus*. Interestingly, all benzimidazole derived silver complexes demonstrated a potential antibacterial activity against *E. coli* with a MIC of 8 μgmL^−1^.

Yildirim et al. functionalized NHC ring tethering a morpholinoethyl moiety [73]. The morpholine ring is present in many antibiotics such as linezolid used for treating infections caused by Gram-positive bacteria, and other derivatives are tested, such as antibacterial [89]. The NHC salts were functionalized with three N-alkyl groups with different steric hindrance and electro-donating and H-bonding properties. The three Ag(I)–NHC complexes (**251**–**253**) were screened against *E. coli*, *S. aureus* and *C. albicans* with a MIC ranging from 5.85 to 44.4 μgmL^−1^. These values were comparable with those recorded treating with imidazole salts. The lowest MIC values were found against bacteria strains for 3-hydroxypropyl substituted imidazole ring whereas the bulkiest imidazole ring showed the best activity *C. albicans*. These outcomes suggest that the alkyl groups, which contain bulky (isopropyl), electron-donating (propyl) and H-bonding (hydroxypropyl) wingtip on the imidazole ring, may play an important role in the antimicrobial activity.

Aher et al. tested morpholine containing silver complexes tuning the side chain to evaluate the possible changes in antimicrobial effects. (**254**–**258**) [74]. The compounds **256** and **255** with butyl and methyl side chain in addition to the morpholine moiety more than other compounds inhibited the bacterial growth with MIC of 12.5 and 6.25 µmolL^−1^ against *S. aureus*, respectively, and 50 µM against *S. enterica* with an efficient growth inhibition up to 21 days with disruption of cell wall as monitored by SEM analysis. The Ag–NHC had shown enhanced bactericidal effects due to the synergistic effects of NaCl which increase solubility product for anionic AgCl_2_^−^.

Recently Belhi et al. synthesized four novel morpholinethyl Ag(I)–NHC complexes (**259**–**262**) [90]. The MIC of the 1,3-dialkyl-5,6-dimethyl-benzimidazolium salts and their complexes was determined for *E. coli*, *P. aeruginosa*, *S. aureus*, *C. glabrata* and *C. albicans* in vitro through BMD (Broth Microdilution). Although all Ag(I)-NHC complexes were active, the complex **262** exhibited a significant broad-spectrum antimicrobial activity.

Patil et al. selected 1,3–benzoxazole and 1,3–dioxolane as substituent on N atom to a series of NHC–precursors and their corresponding Ag(I) complexes [75]. Both 1,3–benzoxazole and 1,3–dioxolane derivatives have been previously studied for their pharmaceutical role as moiety present in wide number of drugs. Four dioxolane **263**–**266** and four benzoxazole **267**–**270** derivatives silver carbene were synthesized. The benzoxazole group in NHC-Ag(I)–NHC was able to coordinate with his nitrogen another Ag cation producing a binuclear silver complex (**269**–**270**). This structure was confirmed by X-ray crystallography. Antimicrobial properties were evaluated against four Gram-negative bacteria (*E. coli*, *K. pneumoniae*, *A. baumannii* and *P. aeruginosa*), one Gram-positive bacteria (*S. aureus*) and two fungi (*C. albicans* and *Cryptococcus neoformans*). The benzimidazolium hexafluorophosphate salts and the silver(I) complexes showed comparable antimicrobial activity in the range 0.2–26.90% of bacterial growth inhibition and fungal growth inhibition in the range 126.30–49.70%. However, the 1,3–dioxolane Ag(I)-NHC acetate **263**–**264** complexes achieved extraordinarily growth inhibition of all the bacterial (>95%) and fungi (>97) strains.

More recently, Lasmari et al. synthesized six 1,3–dioxolane NHC derivatives functionalizing the benzoimidazole ring with substituted benzyl (**271**–**276**) [76]. All complexes were screened for their antibacterial, antifungal and anti-cholinesterase activities exhibiting moderate antibacterial and antifungal activities, generally more active than corresponding NHC salts. The best inhibitor of bacterial growth was found the compound **275** bearing the anthracen-9-ylmethyl functionalized ligand with a value of MIC ranging between 6.25–12.5 μgmL^−1^. Furthermore, the complex **271** was 2 times more active against *P. aeruginosa* than the standard drug Ampicillin with MIC value 6.25 μgmL^−1^. The compounds **272** and **273** gave the best inhibitory activity against *C. albicans* fungi with MIC values 12.5–25 μgmL^−1^. The results proved that the compounds indicated moderate to excellent activity against both acetylcholinesterase (AChE) and butyrylcholinesterase (BuChE). It was found that most of the complexes displayed good AChE and BuChE inhibitory activities. In particular, **273** and **275** complexes were the most powerful inhibitors with IC_50_ values of 8.56 ± 1.17 μM and 5.05 ± 0.30 μM against AChE and BChE, respectively. Docking studies exposed that these compounds bind manly to the catalytic anionic site (CAS) of the AChE and BChE, respectively.

**Figure 12 molecules-28-04435-f012:**
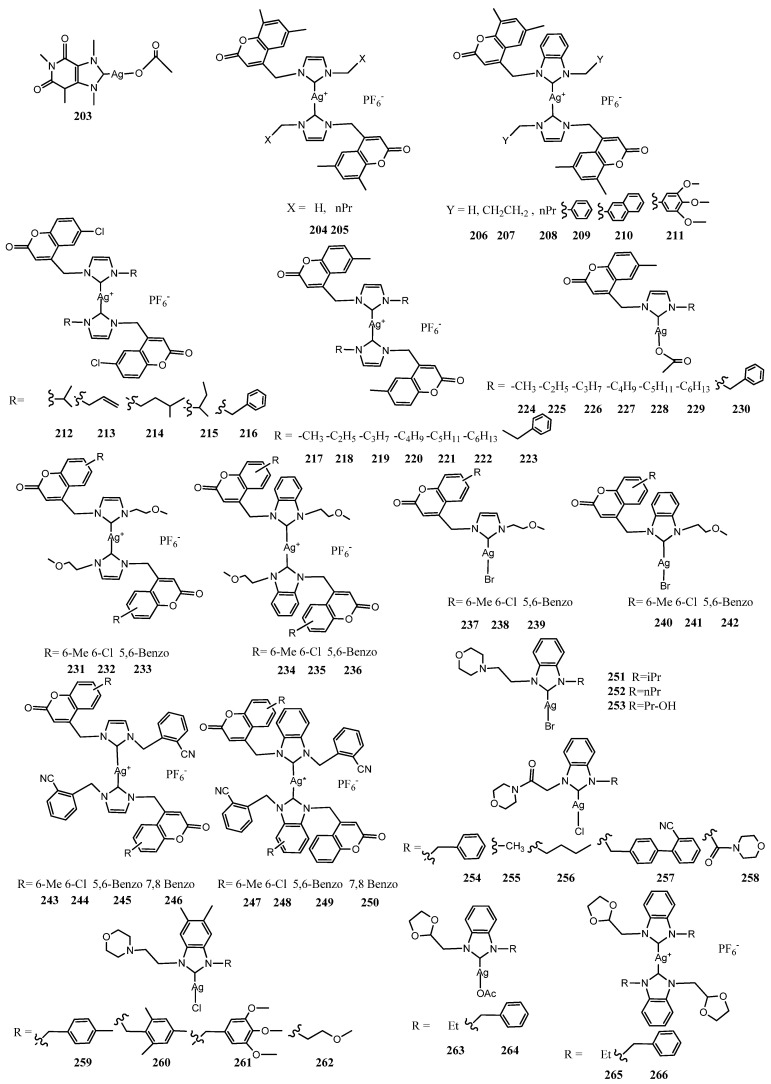
Structures of NHC–silver complexes coordinated by ligands with dual activity.

## 5. Binuclear NHC–Silver Complexes

In the last decade, ligands bearing two imidazole moieties were designed to coordinate two metal center. These complexes should exhibit higher effectiveness than mononuclear complexes due to the presence of two metal center. All structures of reviewed complexes in this section are in Figure 13 and Figure 14 and the antimicrobial data of the most active inhibitor are in Table 2. In a first attempt, Haque et al. synthesized a binuclear silver(I) complexes (**277**–**280**) of allyl and/or alkyl substituted imidazole core of NHCs [77]. The imidazole rings were linked through p-xylyl/2,6-lutidinyl group. The complexes were tested against *E. coli* and *S. aureus* bacterial strains in vitro. The complexes bearing pentyl and hexyl substitution on imidazole (**278**–**279**), have the major antibacterial activity than propagyl substitution (**277**) with 25.6 ± 0.5, 27.0 ± 0.7 and 26.0 ± 0.8, 27.5 ± 2.0 mm of ZoI against both tested bacteria, respectively with 5 μL solution of 100 μgmL^−1^ concentration. The authors compared to a series of mono-nuclear complexes (**281**–**283**) [91] exhibiting threefold lesser activity at the same biological conditions against the same bacterial strains. The enhanced activity of binuclear complexes could be ascribed to the higher stability of the complexes due to the presence of two metal centers. This event allows the slow release of silver ions at required sites.

The higher activity of binuclear complexes was demonstrated by Asekunowo et al. [53]. The binuclear Ag(I)-NHC complex **284** complex was compared to **137**–**139** showing a better binding ability for DNA (Kb = 3.627 × 10^6^). Furthermore, the complex **284** was the most active among the tested complexes on mosquitocidal activity against the dengue vector *Aedes albopictus* affecting the developmental progression from larvae to adult stage. Moreover, binuclear complex **284** outperformed the rest, with the lowest MIC value (12.5 μgmL^−1^) against both *E. coli* and *S. aureus*.

In 2016, Nomiya et al. reported the synthesis, characterization and the structure–activity relationship of three dinuclear Ag(I)–NHC complexes (**285**–**287**) [78]. Each of the three complexes contains four imidazole rings that are linked by either an *ortho*-xylene or a *para*-biphenyl, which induce stability to the final dinuclear silver complexes in both solution and solid state. The antibacterial activity was not as high as expected. Indeed, the binuclear complexes were evaluated against two Gram-negative bacteria (*E. coli* and *P. aeruginosa*) and two Gram-positive bacteria (*S. aureus* and *B. subtitis*), as well as four fungi (*C. albicans*, *S. cerevisiae*, *A. brasiliensis [niger]* and *P. citrium*), with MIC ranging from 15.7 to >1000 μgmL^−1^. The most effective binuclear Ag(I)-NHC compound **285** was the complex with a planar central spacer unit with MIC ranging from 15.7 to 125 μgmL^−1^. Therefore, the activity is dependent on the molecular structure suggesting a SAR.

In the same year, Rizali and co-workers reported six binuclear Ag(I)–NHC complexes liked through an *n*-butyl chain with disappointing biological results, (**288**–**293**) [79]. Indeed all complexes screened against *E. coli* and *S. aureus*, showed activity minor the ampicillin, observing ZoI for *E. coli* and *S. aureus* was approximately 8 ± 1 mm using 9 μL of solution.

The same group designed eight more binuclear Ag(I)–NHC complexes **294**–**301**, shorting the *n*-butyl chain to an ethyl chain and using *N*-alkyl chain to functionalize instead of *N*-benzyl group to improve the effectiveness [80]. In this case, these small variations induce only a slight increase in the antibacterial activity of the complexes compared to the previously studied compounds. For instance, complex **288** exhibited ZoI of 7 ± 1 mm versus 11 ± 1 mm for complex **298**, when screened against *E. coli*.

In the same year, Rizali and coworkers described the synthesis of three more binuclear Ag(I)–NHC complexes (**302**–**304**) [81]. The effect of the length of the alkyl linker chain was investigated, using either an ethyl, propyl or butyl chain. The antibacterial activities against *E. coli* and *S. aureus* showed a decreasing pattern with the increasing of the chain length in the bridging subunit. The complex **302** exhibited against both strains a lower MIC value (12.5 μM). This indicates that longer alkyl chain results in less stable complexes probably due to the weaker argentophilic interaction.

Iqbal and coworkers designed binuclear Ag(I)–NHC (**305**–**307**) linked with a longer or branched alkyl chain [82]. The bis Ag(I)–NHC complex in the test in vitro against three bacterial strains (*B. subtillis*, *B. cereus* and *M. brunensis*) showed an increase in the ZoI with values ranging ZoI of 17 ± 1 mm–28 ± 1 mm and MIC 15 ± 0.1–27 ± 1 values compared to their respective ligands as well as standard drug Ciprofloxacin. In general, complexes were more effective against *B. cereus* as compared to *B.subtillis* and *M.brunensis*. All the compounds presented low hemolytic activity (0.51–8.09%) toward erythrocytes therefore they can be employed in preclinical trials resulting no-hazardous for mouse blood cells. Subsequently, expanding the same class, alkyl linkers were substituted with an aryl link [83]. Both synthesized bis Ag(I)–NHC complexes (**308**–**309**) were tested against three bacterial strains (*S. aureus* and *S. pneumonia* and *E. coli*) finding the MIC value in the range of 2.09–2.35 µg mL^−1^ which was almost in level to broad spectrum standard drug ciprofloxacin and very much better than silver sulfadiazine. More recently, two aryl-substituted bis-benzimidazole binuclear Ag(I)–NHC complexes (**310**–**311**) were described [84]. Both complexes showed ZoI for the Gram-positive (*S. Aureus*) and Gram-negative (*E. coli*) strains, i.e., 18.67  ±  0.58 and 19.83  ±  0.29 mm and 12.5  ±  0.5 and 14.33  ±  0.58 mm for *E. coli*, respectively. Moreover, they show higher ability than their respective salts to inhibit the growth of bacteria at low concentration 0.765 and 0.382 μgmL^−1^ for *S. Aureus* and 3.06 and 1.53 μgmL^−1^ for *E. coli*.

Hussaini et al. synthesized binuclear Ag(I)–NHC (**312**–**316**)) bearing n-alkyl homologous series with propylene spacers to investigate the effect of lipophilicity [85]. The Ag(I)-NHC complexes exhibit activities with ZoI of 11 ± 1–20 ± 1mm and 9 ± 0–16 ± 1mm using 12 μL of solution against *E. coli* and *S. aureus* respectively. In comparative studies, it was observed that the lipophilicity and biological activities of complexes are in positive correlation for both bacterial strains.

Umie et al. reported two bisbenzimidazol2-ylidenes Ag(I)–NHC and eight open chain tetrabenzimidazol-2-ylidenes Ag(I)–NHC complexes (**317**–**326**) [86]. All complexes exhibited against *E. coli* (ATCC 25,922) and *S. aureus* (ATCC 12,600) medium to higher ZoI compared to the standard antibiotic, Ampicillin, without enormous differences. There were no obvious difference between the activity of Ag(I)-NHC complexes compared to the silver-tetra-NHC complexes bearing both two Ag^+^ cations per molecule. Compared to the mononuclear Ag(I)–NHC complexes, they displayed higher activities due to the higher number of Ag^+^ per molecule, as previously demonstrated. Based on obtained data, *E. coli* depicted the highest resistance against **323** while **322** shows the highest antibacterial activity against *S. aureus*.

**Figure 13 molecules-28-04435-f013:**
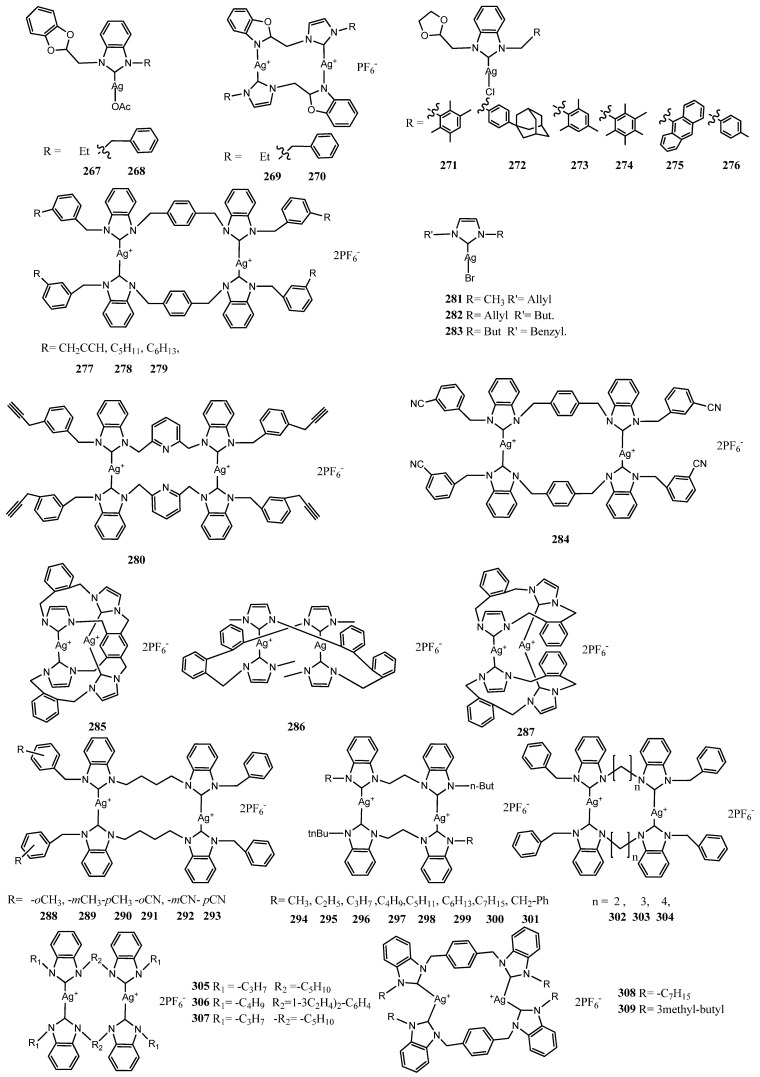
Structures of binuclear NHC–silver complexes.

**Figure 14 molecules-28-04435-f014:**
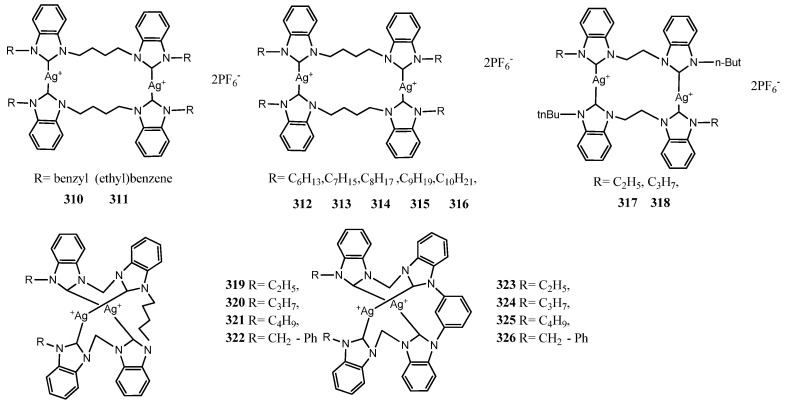
Structures of binuclear NHC–silver complexes.

## 6. Encapsulated Nanosystems Silver–NHC Complexes

The controlled and specific release of drugs plays a pivotal role in different medical applications, often avoiding harmful high concentrations and allowing chemicals to carry out their task for a prolonged period. Targeted release of silver cations at infection site for application in vivo is crucial. Despite this consideration, few publications are reported until today. Although the promise results were obtained in vitro against different strains of bacteria, two concerns may arise: the presence of chloride anions and sulfur containing proteins in the bloodstream. The chloride anions may in some cases improve the bioavailability of Ag forming AgCl_2_^−^ more soluble than silver chloride, but in other cases destabilize the complexes favoring the silver chloride precipitation. The thiol group of cysteine can bind the silver preventing to reach the site of infection and macrophages that capture the Ag(I)–NHC. All of these drawbacks can be overcome by encapsulating the complexes in nanosystems. Pulmonary drug delivery is one of the goals. For this purpose, Young and coworkers formulated L-tyrosine polyphosphate nanoparticles LTP NPs [92] (Figure 15). Although the **327** Ag(I)–NHC displayed very low MIC (1 μgmL^−1^) against *P aeruginosa* and *E. coli* **327**-loaded into LTP NPs still exhibits excellent antimicrobial activity in vitro and in vivo against chronically infected lungs of cystic fibrosis (CF) relevant bacteria *P aeruginosa* with an MIC value of 4 μgmL^−1^. These nanosystems provide sustained release of the Ag^+^ cations over the course of several days providing significant survival advantage in mouse infection models with only two doses. Surfactant and polymer micelles have been demonstrated to be an excellent system to deliver hydrophobic drugs forward a precision medicine [93]. The same authors exploited this opportunity [94]. They encapsulated the **327** compound and silver nitrate into the hydrophobic core domain of micelles obtained assembling amphiphilic block copolymers, poly(acrylic acid)-*b*-polystyrene (PAA-*b*-PS). These systems release 50% of silver within ca. 1 day and ca. 80% within 2 days and achieved a plateau with full release by ca. 4 days, a period that would provide a desired effect for therapeutic delivery. Moreover, degradable acetalated dextran (Ac-DEX) nanoparticles were prepared and loaded by a single-emulsion process with **328 [95]**. The bacterial growth kinetics, linked to release silver kinetics from the particles, demonstrated that Ac-DEX- nanoparticle formulations were active against all bacterial strains including the silver-resistant strain, *E. coli J53* + *pMG101*. Compared to the free drug, the Ac-DEX nanoparticles were much more easily suspended in an aqueous phase and subsequently aerosolized, thus providing an effective method for pulmonary drug delivery.

A series of anionic dNPs from block copolymers having polyphosphoester (PPE) and poly(l-lactide) (PLLA) block segment were designed and formulated specifically for silver loading into the hydrophilic shell and/or the hydrophobic core as potential delivery carriers for silver acetate or **327** and **135** for applications in direct epithelial treatment and in urinary tract [96]. The complexes were loaded following three different ways: (1) electrostatic interaction with carboxylate groups within the hydrophilic corona; (2) coordination with the two sulfur atoms of the 1,2-dithioether moieties on the side chains of the PPE block segment; and (3) encapsulation by hydrophobic interactions with the hydrophobic PLLA core of the nanoparticle (Figure 15).

The release kinetics of silver-bearing dNPs revealed 50% release at ca. 2.5–5.5 h depending on the type of silver compound. The Ag(I)–NHC in the dNP-based delivery system improved MIC up to 70%, compared with the free complexes, as detected in vitro against 10 contemporary epidemic strains of *S. aureus* and eight uropathogenic strains of *E. coli*.

**Figure 15 molecules-28-04435-f015:**
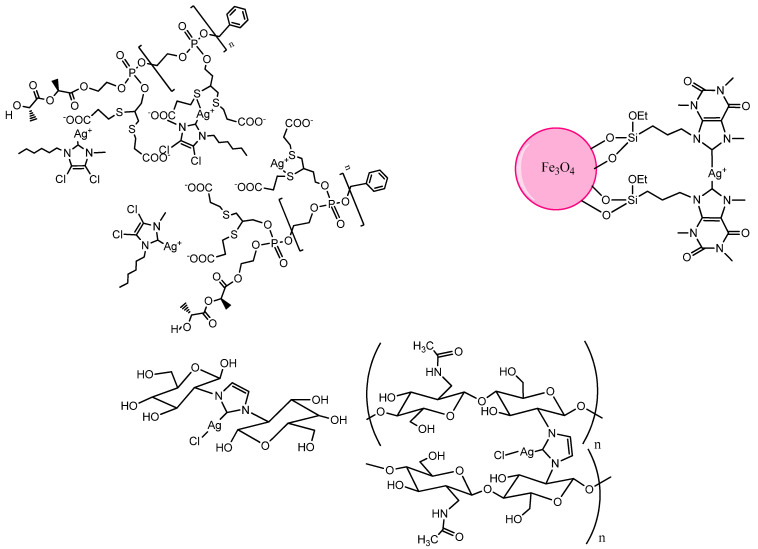
Structures of encapsulated NHC–Silver complexes.

More recently, Necol et al. selected chitooligosaccharides (COS) and its monomer *D*-glucosamine to bind Ag^+^ through the NHC bond (Figure 15) [97].

The deacetylation of the starting material is a limiting factor for the synthesis of NHC-Ag-COS, since free amino groups are required for the synthesis of the NHC and consequent introduction of silver. The binding silver to chitosan oligomers and monomer holds and eventually improves the activity of silver in vitro. The activities were compared with that of silver nitrate and the antiseptic polyvinylpyrrolidone iodine (PVP iodine), carrying the active iodine. Based on these results, the use of NHC chitooligosaccharides is a promising approach for improving silver selectivity, achieving a consequent reduction in its toxicity and thus providing better silver formulations for clinical practice.

Very recently a Caffeine based Ag(I)–NHC complex anchored on magnetic nanoparticles (MNP-Caff-NHC@Ag complex) has been prepared by covalent grafting of caffeine on the surface of chloro-functionalized Fe_3_O_4_ magnetic nanoparticles followed by complexation with silver (I) acetate (Figure 15) [98]. The MNP-Caff-NHC@Ag complex displayed significant antibacterial activity against *E. coli* (NCIM-2832), *S. aureus* (NCIM-2654) and *B. cereus* (NCIM-2703) with zone of clearance between 7 mm and 17 mm.

## 7. Perspective and Conclusions

After the discovery of penicillin, it seemed that microbial infections would have been outdated. However, the emergence of the resistance to antibiotics has revived the fight against these pathologies. Silver can play a pivotal role in this field. Bioavailability and delivery of silver cations as active specie is the goal of scientists. The exploration of silver complexes is a valid alternative to silver nanoparticles. Complexes may enhance the efficiency by releasing the metal ions into the targeted area of the cell membrane and retain the activity over a long period. Many silver complexes were tested in medicinal applications, but the vast majority contained the NHC system. Some examples of carboxyl complexes, as well as nitrogen complexes, are reported in the literature [99,100]. Nomiya et al. [99] suggested a mechanism for the antimicrobial activity, mainly determined by solubility, silver transport phenomena and ligand exchange equilibria rather than carboxylic acids coordination. Therefore, NHC ligands were privileged for their versatility, synthetic accessibility and stability.

More of three hundred complexes of NHCs were designed and tested in the last twenty years. The factors affecting antimicrobial activity are identified in Figure 16 as most critical (green), relevant (yellow) and least effective (red).

All achievements agree that complexes must meet two requirements: they should retain their ligands releasing silver cation over prolonged time and ligands should guarantee sufficient lipophilicity. As a lipid membrane surrounds the microbial cell wall, which favors the pathway of fat-soluble materials, the increase in lipophilicity allows penetration of the complex into and across the membrane. Inside the cell they deactivate the active enzyme sites of the microorganisms.

The structure–activity relationships (SAR) are related to the substituents on imidazole or benzoimidazole rings at all positions. Electronic stability and steric hindrance of NHC stabilize the metal-NHC complexes for better bioactivity. The strength of Ag–NHC bond is influenced by the presence of substituent on 4 and 5 imidazole position. However, the presence of electron donor or withdrawal group does not induce significant variations on effectiveness. The lipophilicity can be fine-tuned by a structural versatile substituent attached to the benzimidazole or imidazole ring. Nitrogen’s substituents as aryl and alkyl chain length are known to have a noteworthy effect on the bioactivity. All reported assays are in agreement that the amount of complex penetrating into the lipid bilayer is directly proportional to the increase in ligand carbon chain length. Furthermore, the presence of aromatic ring of benzimidazole further enhances their activity due to increased lipophilicity disrupting the function of cellular organelles which consequently results in the obstruction of respiratory and metabolic mechanisms of microbes. Although benzimidazole derivative molecules effectively interact with microbial targets, these ligands seem not play a specific role in antibacterial activity.

After membrane penetration, broad antimicrobial spectrum of Ag(I)–NHC complexes with large number of targets might be attributed to ligand exchange phenomenon with S-(thiols), N-or O-donors at their potential target points.

Therefore, in last years, molecular docking analysis were carried out to predict possible interactions of silver complexes with specific protein targets. These studies provide both detailed analysis and foresight about the designing of new molecules. The interaction with proteins such as AHL Lactonase, Malate Synthase with the catalytic anionic site (CAS) of the AChE, and BChE can be tested to screen active compounds. However few X-ray or ESI-MS studies are reported on interaction between these targets and Ag(I)–NHC.

Almost all studies confirmed the effectiveness based on silver, whereas azolium salts are inactive. Although many ligands bearing naphthalene or anthracene group are able to intercalate between DNA base, they are inactive, and the efficiency is attributed to silver cation. Moreover, ligands moiety which play pharmacological role such as coumarin or morpholine seemed not significantly increase the antimicrobial properties.

As the type of bacterial strains affects antimicrobial activity in addition to the nature of silver complexes, so general conclusions about the structure–activity relationship to display antimicrobial potential cannot be drawn.

Although few example of silver complexes encapsulation in supramolecular aggregates are reported, this technology can help application in vivo. The high lipophilicity of silver complexes allows polymeric micelles to be loaded into hydrophobic compartment Moreover, these systems can protect the integrity of the complexes avoiding interaction with other proteins in vivo. They allow a slow controlled release to care wounds prolonging the effect of the drug. These achievements are promising for future applications in the fight against microorganism.

## Figures and Tables

**Figure 1 molecules-28-04435-f001:**
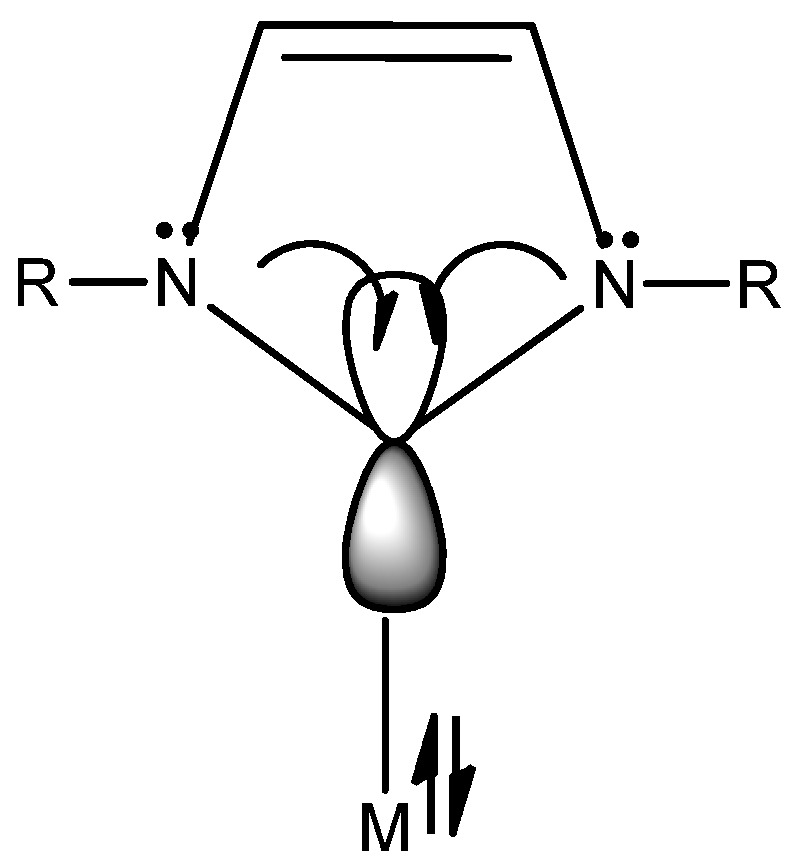
σ donor character of the carbene C2 due to the two adjacent nitrogen lone electron pairs to the free p-orbital.

**Figure 2 molecules-28-04435-f002:**
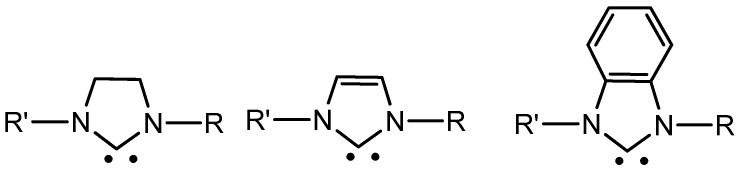
Structure of imidazolidin-2-ylidene (**left**), imidazol-2-ylidene (**center**) and benzimidazol-2-ylidene (**right**).

**Figure 16 molecules-28-04435-f016:**
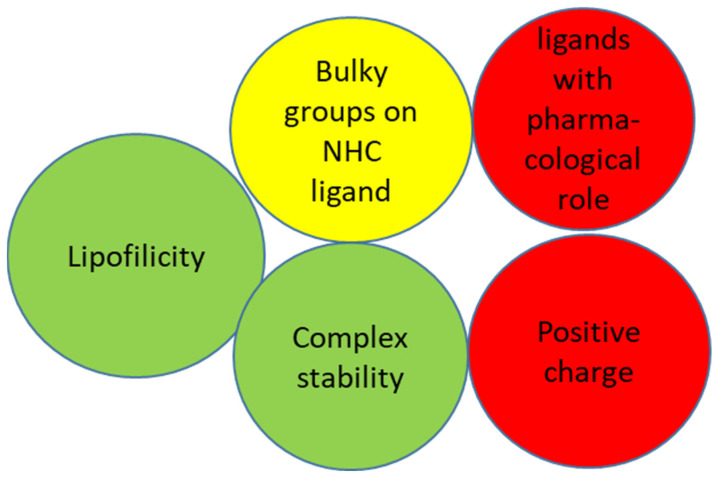
In green: most critical factors, in yellow: the relevant factors and in red: the least effective factors affecting antimicrobial activity.

**Table 2 molecules-28-04435-t002:** Best dual target and binuclear NHC silver complexes inhibitors of each reference.

Entry	N° Strains	Highest Activity Against	Concentration	ZoI or MIC or % of G *	Ref.
**203**	36	*E. coli*–*Burkholderia*	1 μg mL^−1^	MIC	[68]
**210**	6	*E. faecalis*–*S. aureus*	25 μgmL^−1^	MIC	[69]
**210**	6	*C. albicans*–*C tropicalis*	25 μgmL^−1^	MIC	[69]
**212**–**216**	6	*P. aeruginosa*	8 μgmL^−1^	MIC	[70]
**212**	6	*P. aeruginosa*	24 mm at 12 μL	ZoI	[70]
**217, 220**	6	*P. aeruginosa*	8 μgmL^−1^	MIC	[71]
**231**–**236**	4	*S. aureus*–*E. coli*	16 μgmL^−1^	MIC	[72]
**243**–**245**	2	*E. coli*	8 μgmL^−1^	MIC	[60]
**253**	3	*S. aureus*–*E. coli*	5.85 μgmL^−1^	MIC	[73]
**255**	2	*S. aureus*	6.25 µmolL^−1^	MIC	[74]
**263**–**264**	7	*All Strains*	>95%	% of G	[75]
**271, 275**	5	*E. coli*	6.25 μgmL^−1^	MIC	[76]
**271**	5	*P. aeruginosa*	6.25 μgmL^−1^	MIC	[76]
**278**–**279**	2	*E. coli*–*S. aureus*	27.0 mm at 100μL mL^−1^	ZoI	[77]
**284**	2	*E. coli*–*S. aureus*	12.5 μgmL^−1^	MIC	[53]
**285**	8	*S. aureus*–*B. subtilis*	15.7 μgmL^−1^	MIC	[78]
**288**–**293**	2	*E. coli and S. aureus*	8 ± 1 mm at 9 μL	ZoI	[79]
**295**–**296**	2	*E. coli and S. aureus*	13 ± 0.6 mm at 6 μL	ZoI	[80]
**302**	2	*E. coli*–*S. aureus*	12.5 µM	MIC	[81]
**305**	4	*M. brunensis*	15 ± 0.9 μgmL^−1^	MIC	[82]
**308**	3	*S. aureus*	2.09 µM	MIC	[83]
**310**	2	*S. aureus*	0.382 μgmL^−1^	MIC	[84]
**316**	2	*S. aureus*	20 ± 1 mm at 12 μL	ZoI	[85]
**317**	2	*E. coli*	1.56 μM	MIC	[86]
**322**	2	*S. aureus*	20 ± 1 mm at 12 μL	ZoI	[86]

* Percentage of growth.

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
