# Peer review of "Structure–Activity Relationships in NHC–Silver Complexes as Antimicrobial Agents"

_molecules, 2023, doi:10.3390/molecules28114435_

Round 1

Reviewer 1 Report

The manuscript describes Ag(I) complexes with N-Heterocyclic carbenes. The authors have done a well done job about structure–activity highlighting lots of Ag(I) complexes structures. From my point of view the paper is apropriate to be published in the present form.  

Author Response

Reviewer 1

The manuscript describes Ag(I) complexes with N-Heterocyclic carbenes. The authors have done a well done job about structure–activity highlighting lots of Ag(I) complexes structures. From my point of view the paper is apropriate to be published in the present form.  

We appreciate the reviewer comments

Reviewer 2 Report

Major comments

This is a valuable manuscript which thoroughly reviews important content on a large variety of silver complexes that have antimicrobial activity. As the review is extensive, some more subheadings, a list of content on the first page and a summarizing and guiding schematic figure on structure-activity relationships might be considered. Also some textual shortening might be considered.

Minor comments

L10: are with “silver” silver ions, silver or both indicated?

L11: the major drawback of silver salt is the limited duration of its antimicrobial activity

L12: the difference between silver complexes and nanoparticles is not obvious without definition/description.

L13: Please is the following meant: broad-spectrum silver-containing antimicrobial agents are well represented by N-Heterocyclic carbene (NHC) silver complexes.

L14: this class of complexes

L15: Please enhance clarity: introducing different substituents on – 15 N-Heterocycle

L17: Gram-positive, Gram-negative

L18: underlining or underlying

L20: in polymer-based supramolecular aggregates are reported (no “a” before polymer)

L21: Please rephrase for clarity: This point represents the most promising frontiers for the targeted delivery of silver ions to the infected sites.

L27: issues

L28-29 not clear

L30: metal-based

L33: organisms.

L33: Nevertheless, Ag+

L52: it was found

L53: cation concentration

L54: Please rephrase for clarity: moreover increase sliver activity

L58: bacterial cell membranes

L60: inside the cell,

L61: enzymes

L62: of these mechanisms

L63: Moreover,

L65: binds

L66: Therefore,

L67: identified.

L67: E. coli

L69: [13].

L70: ways

L84-86: unclear

L92: centers

L96-98: suggested to add a figure to clarify

L98: phosphines [19].

L106: the most common types are

L176: [29].

Table 1: of each reference or of each strain? Please explain the methods of ZoI, A of Cl and MiC in a legend

L853: The authors declare no conflict of interest.

L849: discovery.

References: numerous tiny formatting issues , e.g. 10: Duran, N.;

Constructively, it is recommended to slightly enhance the sometimes heterogenous textual clarity and to use a spell check program, to optimally make this manuscript of interest to a broad range of readers.

Author Response

Reviewer 2

Major comments

This is a valuable manuscript which thoroughly reviews important content on a large variety of silver complexes that have antimicrobial activity. As the review is extensive, some more subheadings, a list of content on the first page and a summarizing and guiding schematic figure on structure-activity relationships might be considered. Also some textual shortening might be considered.

We introduced subheadings in the third paragraph to highlight the structure of NHC designed.   We did not reported a list of contents because “Molecules” does not provide for the possibility of reporting paragraph lists on the first page.

Minor comments

L10: are with “silver” silver ions, silver or both indicated?

We indicate with silver, the use of the metal since the ancient era. In the last century it was demonstrated that the active specie is Ag+ cation.

L11: the major drawback of silver salt is the limited duration of its antimicrobial activity

We got the suggestion

L12: the difference between silver complexes and nanoparticles is not obvious without definition/description.

We remove this sentence from the abstract. It lacks enough characters to explain the meaning of this sentence

L13: Please is the following meant: broad-spectrum silver-containing antimicrobial agents are well represented by N-Heterocyclic carbene (NHC) silver complexes.

We got the suggestion

L14: this class of complexes

We got the suggestion

L15: Please enhance clarity: introducing different substituents on N-Heterocycle

We modified the sentence: “The properties of NHC can be tuned introducing side chains on N-Heterocycle”

L17: Gram-positive, Gram-negative

We got the suggestion

L18: underlining or underlying

underlining

L20: in polymer-based supramolecular aggregates are reported (no “a” before polymer)

We got the suggestion

L21: Please rephrase for clarity: This point represents the most promising frontiers for the targeted delivery of silver ions to the infected sites.

We modified the sentence: Delivery of silver complexes to the infected sites will be the most promising goal for the future.

L27: issues

We got the suggestion

L28-29 not clear

L30: metal-based

We got the suggestion

L33: organisms.

We got the suggestion

L33: Nevertheless, Ag+

We got the suggestion

L52: it was found

We got the suggestion

L53: cation concentration

We got the suggestion

L54: Please rephrase for clarity: moreover increase sliver activity

We removed this sentence.

L58: bacterial cell membranes

We got the suggestion

L60: inside the cell,

We got the suggestion

L61: enzymes

We got the suggestion

L62: of these mechanisms

We got the suggestion

L63: Moreover,

We got the suggestion

L65: binds

We got the suggestion

L66: Therefore,

We got the suggestion

L67: identified.

We got the suggestion

L67: E. coli

We got the suggestion

L69: [13].

We got the suggestion

L70: ways

We got the suggestion

L84-86: unclear

We modified the sentence: “Since their discovery by Arduengo et al. [16] NHC have become universal ligands in organometallic and inorganic coordination to transition metals in catalytic applications [17] for its versatile properties”.

L92: centers

We got the suggestion

L96-98: suggested to add a figure to clarify

We added a figure to clarify the concept

L98: phosphines [19].

We got the suggestion

L106: the most common types are

We got the suggestion

L176: [29].

We got the suggestion

Table 1: of each reference or of each strain? Please explain the methods of ZoI, A of Cl and MiC in a legend

We reported in the table the complexes with the best antimicrobial activity for each reference.

L853: The authors declare no conflict of interest.

We got the suggestion

L849: discovery.

We got the suggestion

References: numerous tiny formatting issues , e.g. 10: Duran, N.;

We checked formatting issues of all references

Reviewer 3 Report

This review is focused on biologically active Ag(I) complexes supported by N-Heterocyclic carbenes (NHCs). Over the last decade, these compounds have attracted the attention of many scientific groups owing to their stability and potential application as antimicrobial agents. The review is conveniently structured, and for each compound class, the information regarding the preparation, emission properties, and possible applications is discussed. In general, the authors managed to maintain the necessary balance between detailed descriptions and generalizations. In my opinion, this review will be very useful both for specialists working in the field of bioinorganic chemistry and for the general community dealing with Ag(I) complexes. I recommend addressing the following issues before acceptance of the manuscript:  
1. Some of the depicted structures are depicted incorrectly. Please, check carefully all the drawings. 
2. Authors should compare the NHC-based Ag(I) complexes with other Ag(I) complexes in the terms of antimicrobial activity. 
3. In fact, the "Structure–Activity Relationships" didn't deeply discus in the review, although it's in the title of the MS. Please pay more attention to this important aspect. Without discussing these relationships, the review looks like a simple compilation of published data.

Author Response

Reviewer 3

This review is focused on biologically active Ag(I) complexes supported by N-Heterocyclic carbenes (NHCs). Over the last decade, these compounds have attracted the attention of many scientific groups owing to their stability and potential application as antimicrobial agents. The review is conveniently structured, and for each compound class, the information regarding the preparation, emission properties, and possible applications is discussed. In general, the authors managed to maintain the necessary balance between detailed descriptions and generalizations. In my opinion, this review will be very useful both for specialists working in the field of bioinorganic chemistry and for the general community dealing with Ag(I) complexes. I recommend addressing the following issues before acceptance of the manuscript:  

  1. Some of the depicted structures are depicted incorrectly. Please, check carefully all the drawings.

We thanks the reviewer. We proceeded to a careful revision of all figures.

  1. Authors should compare the NHC-based Ag(I) complexes with other Ag(I) complexes in the terms of antimicrobial activity. 

The vast majority of silver compounds are NHC based. There are in literature other types of ligands, such as carboxylate products or N -bonds containing silver(I) complexes​. Most of these other complexes were tested for their anticancer properties. Only few examples of complexes were reported with activity against bacteria and fungi strains. In the conclusion we reported some considerations about this class of compounds.

  1. In fact, the "Structure–Activity Relationships" didn't deeply discus in the review, although it's in the title of the MS. Please pay more attention to this important aspect. Without discussing these relationships, the review looks like a simple compilation of published data.

In the review the attempts to design NHC complexes and the achieved results are reported The SAR are discussed in the text highlighting in single study the conclusion of the authors. General remarks are reported in the paragraph “Perspective and conclusion”. The most relevant points are lipophilicity and stability of the complexes whereas the bulky substituents on imidazole and benzoimidazole rings are less relevant to increase the antimicrobial activity. Moreover, the presence of a positive charge on the complexes is less significant than the stability inducted by ligands. The main achievements on structure-activity relationships are now summarized in the figure 10

Reviewer 4 Report

The submission by Rongo et al is a review of silver NHC complexs as antimicrobial agents, and purports to focus on structure-activity relationships, SAR. Silver ions have long been know to have antimicrobial activity, and the use of silver NHC complexes  an active area of research, well worth a good review and a suitable area for publication in Molecules.

That said, the manuscript is difficult to read and far from publishable. There are problems in English usage, syntax, word choice, punctuation, sentence structure throughout the text. Problems with spacing in units, number sense, capitalization (N-Heterocylic carbenes!)It needs a thorough rewrite and editing.  Beyond that, there is much that can be done to improve the review, especially in its goal to provide SAR analysis.

The organization of the review is not bad, with sections on mononuclear NHC complexes, ones with dual activity, dinuclear complexes and encapsulated complexes. But the presentation and comparisons complexes are very anecdotal and difficult to get through. SAR comparisons at conclusion are poorly defined and defended.

The main problem with comparing anti-microbial activity is that it is measured in a variety of different means  (MIC, %G or ZOI/AofCl) and in different units of dose response (micromolar vs microgram per milliliter). Thus attempts to compare activity between papers requires more thorough analysis than is given here. The most reliable way would be for the reviewers to convert the dosage and activity data such that all reports can be compared on a common scale. For MIC this would mean converting mg/mL to molarity - easily done by using the molecular weight of the drug, but that would entail the reviewers to do this conversion for all the compounds described instead of simply reporting the values given in the papers. A more valid comparison would be Ag concentration per dose, [Ag], as this is the active species the complexes are delivering. Likewise, Zone of Inhibition and Area of Clearance are essentially the same measurements, and the reported values would need to be converted to the same units using molarity of dosage. If some reports are not comparable due to methods used, then leave them out!  Choose a cohort of reports that allow you to make real comparisons. I would suggest comparing MIC and ZoI/AofCl in separate tables… e.g., only compare apples to apples. This would allow actual SAR analysis of their effectiveness.

A second problem is that they combine chemical structures into gigantic figures with >50 structures on each!  For SAR, the authors only report the most active species of each report.  I suggest they limit the structures shown to those specific complexes, but discuss variation of structures tested in the  description (much like they already do). 

I also suggest they try to break them into smaller subgroups for figures and discussions. In looking at the structures discussed, they can distinguish between neutral complexes with halide vs carboxylate ligands, and with cationic complexes with bis-NHC ligands (Fig 2 and 3).  They might also separate the bis chelating NHC complexes, the 3 and 4 coordinate Ag complexes, and the polymeric (Fig 3 and 4).  Then compare the most active of each report in a table with common measurements (MIC or ZoI) and units (Ag uM) under each subgroup and compare by effect of [Ag] content between types.  Same type of subgroup comparisons can be made for the dual activity, binuclear and encapsulated complexes. 

There are also a few assumptions that should be clarified.  

Lines 11-12 in the abstract conflate silver salts with silver nanoparticles… is this what you mean to say?  (line 18 is missing the word “bacteria” after Gram negative!!) Abstract needs major rewrite.

Lines 63-67 discuss morphological changes seen in TEM and equate them with misfolded proteins, this needs to be more explicitly described (unfolded protein response generates large vacuoles with in the cells?)

That said, the manuscript is difficult to read and far from publishable. There are problems in English usage, syntax, word choice, punctuation, sentence structure throughout the text. Problems with spacing in units, number sense, capitalization (N-Heterocylic carbenes!)It needs a thorough rewrite and editing. 

Author Response

We appreciate the reviewer comments

Round 2

Reviewer 3 Report

The authors partially took into account the comments of the reviewer. Thus, the review can now be accepted for publication.

Author Response

Thank you for the time and knowledge you have allocated to review our manuscript. We highly appreciate contribution to improve the manuscript of the reviewer.

Reviewer 4 Report

The revision is much improved, both in presentation and in SAR interpretation.  Addition of discussion of the activity of Ag on inner membranes is appreciated, as is added discussion of SAR interpretation.

There remains problems with English usage, most obvious in the newly added text.

line 70 number sense (hypothesis/hypotheses)

73/74 partial repetitive sentence

no paragraph at 76/77, start at 'Indeed'

80 no caps needed

100  'N-heterocyclic carbenes'

109 'NHC-silver'

846 delete 'of silver compounds mainly'

853-7 reword: the factors affecting antimicrobial activity are identified in Figure 15 as most critical (green), relevant (yellow) and least effective (red).

Author Response

Thank you for the time and knowledge you have allocated to review our manuscript. We highly appreciate contribution of the reviewer. We are grateful for pointing out some mistakes in the revised version

Reviewer 4

There remains problems with English usage, most obvious in the newly added text.

line 70 number sense (hypothesis/hypotheses)

We agree with the reviewer it is plural: hypotheses

73/74 partial repetitive sentence

We removed the repetitive sentence

no paragraph at 76/77, start at 'Indeed'

We got the suggestion

80 no caps needed

We got the suggestion

100  'N-heterocyclic carbenes'

We removed the capital letter

109 'NHC-silver'

We removed the capital letter

846 delete 'of silver compounds mainly'

We deleted 'of silver compounds mainly'

853-7 reword: the factors affecting antimicrobial activity are identified in Figure 15 as most critical (green), relevant (yellow) and least effective (red).

We got the suggestion
